# A synthetic optical database generated by radiative transfer simulations in support of studies in ocean optics and optical remote sensing of the global ocean

Hubert Loisel[1], Daniel Schaffer Ferreira Jorge[1], Rick A. Reynolds[2], and Dariusz Stramski[2]

[1]Laboratoire d'Océanologie et de Géosciences, Université du Littoral-Côte-d'Opale, Université Lille, CNRS, IRD, UMR 8187, LOG, 32 avenue Foch, Wimereux, France
[2]Marine Physical Laboratory, Scripps Institution of Oceanography, University of California San Diego, La Jolla, California 92093-0238, USA.

Correspondence: Hubert Loisel (hubert.loisel@univ-littoral.fr)

**Abstract.** Radiative transfer (RT) simulations have long been used to study relationships between the inherent optical properties (IOPs) of seawater and light fields within and leaving the ocean from which the ocean apparent optical properties (AOPs) can be calculated. For example, inverse models to estimate IOPs from ocean color radiometric measurements have been developed or validated using results of RT simulations. Here we describe the development of a new synthetic optical database based on hyperspectral RT simulations across the spectral range from the near-ultraviolet to near-infrared performed with the HydroLight radiative transfer code. The key component of this development was the generation of the synthetic dataset of seawater IOPs that served as input to RT simulations. Compared to similar developments of optical databases in the past, the present dataset of IOPs is characterized by probability distributions of IOPs that are consistent with global distributions representative of vast areas of open ocean pelagic environments and coastal regions covering a broad range of optical water types. The generation of the synthetic data of IOPs associated with particulate and dissolved constituents of seawater was driven largely by an extensive set of field measurements of the phytoplankton absorption coefficient collected in diverse oceanic environments. Overall, the synthetic IOP dataset consists of 3320 combinations of IOPs. Additionally, the pure seawater IOPs were assumed following recent recommendations. The RT simulations were performed using 3320 combinations of input IOPs assuming vertical homogeneity within an infinitely deep ocean. These input IOPs were used in three simulation scenarios associated with assumptions about inelastic radiative processes in the water column (not considered in previous synthetically-generated optical databases) and three simulation scenarios associated with sun zenith angle. Specifically, the simulations were made assuming no inelastic processes, the presence of Raman scattering by water molecules, and the presence of both Raman scattering and fluorescence of chlorophyll-a pigment. Fluorescence of colored dissolved organic matter was omitted from all simulations. For each of these three simulation scenarios, the simulations were made for three sun zenith angles of 0°, 30, and 60° assuming clear skies, standard atmosphere, and wind speed of 5 m s$^{-1}$. Thus, overall 29880 RT simulations were performed. The output results of these simulations include the radiance distributions, plane and scalar irradiances, and the whole set of AOPs including the remote-sensing reflectance, vertical diffuse attenuation coefficients, and mean cosines where all optical variables are reported in the spectral range from 350 to 750 nm at 5 nm intervals for different depths between the sea surface and 50 m. The consistency of this new synthetic database has been assessed through comparisons with in situ data and previously developed empirical relationships involving the IOPs and AOPs. The database is available at Dryad open-access repository of research data (doi:10.6076/D1630T).

41

## 1 Introduction

The investigation of the propagation of natural light in the ocean can be addressed experimentally through in situ measurements and theoretically through numerical simulations of radiative transfer. The understanding of the relationships between the radiometric quantities (i.e., radiance and irradiances) that characterize the light fields within and leaving the ocean and the inherent optical properties (IOPs) of the water column, boundary conditions at the sea surface (i.e., surface illumination conditions and sea state) and at the ocean bottom (i.e., bottom depth and albedo) requires comprehensive datasets of multiple variables acquired over a broad range of environmental conditions. For example, of particular interest are the relationships between the spectral remote-sensing reflectance of the ocean (in sr$^{-1}$), $R_{rs}(\lambda)$, which is an apparent optical property (AOP) derivable from radiometric quantities, and the seawater IOPs that are directly linked to various seawater constituents because these relationships form the cornerstone of various applications of optical (ocean color) remote sensing. Recent technological developments and broader accessibility of optical in situ instrumentation have led to significant increase in optical datasets collected across diverse oceanic environments and efforts have been undertaken to merge data from various sources within publicly available databases (e.g., Werdell and Bailey, 2005; Valente et al., 2019; Casey et al., 2020). Although the importance of field data collection across diverse environments cannot be overstated, the existing database compilations are subject to certain limitations. In addition to typical measurement errors, it is difficult to ensure consistent data quality and characterization of uncertainties across all merged data because individual datasets are often obtained with different instruments as well as measurement and data processing methods. Also, even the large databases such as NASA's SeaWiFS Bio-optical Archive and Storage System (SeaBASS, https://seabass.gsfc.nasa.gov/) cannot ensure the balanced representativeness of collected field data in terms of a broad range of optical conditions across diverse ocean environments. In this context, radiative transfer (RT) simulations, which are free of measurement errors, provide a useful tool to generate comprehensive synthetic databases and complement the existing datasets of field measurements in support of studies in ocean optics and optical remote sensing.

Over the past decades various radiative transfer models that employ different numerical solution techniques have been developed and used to address a wide range of problems related to optics of natural water bodies (e.g., Mobley et al., 1993; Mobley, 1994; Stamnes et al., 2017). Since the 1990s the HydroLight code based on invariant imbedding technique (Mobley, 1989; Mobley et al., 1993; Mobley, 1994) has been among the most commonly used radiative transfer models in oceanographic optics. The HydroLight code solves the scalar (i.e., polarization of light is not included) time-independent radiative transfer equation for a horizontally homogeneous water body, in which the inherent optical properties can vary with depth, under given boundary conditions at the surface and bottom of the water body. The inelastic radiative processes within the water column that include Raman scattering by water molecules, fluorescence of chlorophyll-a pigment, and fluorescence of colored dissolved organic matter (CDOM) can be included in HydroLight simulations.

The radiative transfer simulations with HydroLight code have proven useful for generating synthetic databases of light field characteristics (i.e., radiance and irradiances) within and leaving the ocean and the AOPs derived from the simulated radiometric quantities for various scenarios of seawater IOPs that provide input to the simulations. In particular, as a result of efforts dedicated to inverse bio-optical algorithms and coordinated under

the auspices of the International Ocean Colour Coordinating Group (IOCCG Report, 2006), a widely-used publicly available synthetic database was generated within the spectral range 400 - 800 nm with a 10 nm resolution for clear sky conditions with three different sun zenith angles (0°, 30°, and 60°), a sea surface state corresponding to a wind speed of 5 m s$^{-1}$, and 500 different IOP combinations driven by chlorophyll-a concentration, Chla, within the surface ocean layer. The input IOP data included the spectral absorption coefficients of phytoplankton, $a_{ph}(\lambda)$, non-algal particles (also referred to as depigmented or detrital particles that can include various types of particles such as organic detritus, mineral particles, heterotrophic bacteria, and depigmented phytoplankton cells), $a_d(\lambda)$, colored dissolved organic matter (CDOM), $a_g(\lambda)$, and the spectral backscattering coefficients of phytoplankton, $b_{b-ph}(\lambda)$, and non-algal particles, $b_{b-d}(\lambda)$ ($\lambda$ represents the wavelength of light in vacuum in units of nm and the IOP coefficients are typically expressed in units of m$^{-1}$). The output parameters provided by those simulations that are available in the public database included the following AOPs: the spectral remote-sensing reflectance, $R_{rs}(\lambda)$, the remote-sensing reflectance just below the sea surface, $r_{rs}(\lambda)$, the irradiance reflectance just below the sea surface, $R(z=0^-, \lambda)$, and the diffuse attenuation coefficient for downwelling plane irradiance, $K_d(\lambda, z)$, at the depths $z = 0^-$, 5, and 10 m (where $0^-$ indicates the depth just beneath the sea surface).

Another synthetic database that is publicly available was developed as part of the CoastColour Round Robin project (Nechad et al., 2015). This project was focused on coastal waters and IOPs were described by 5000 combinations of Chla, $a_g(\lambda)$, and mass concentration of mineral particles. The HydroLight simulations were run from 350 nm to 900 nm at 5 nm intervals for cloudless sky, three sun zenith angles (0, 40, and 60°), and a wind speed of 5 m s$^{-1}$. The output parameters included in the publicly available database are the water leaving reflectance, $RL_w(\lambda) = \pi R_{rs}(\lambda)$, $K_d(\lambda)$, the photosynthetically available radiation, $PAR$, and the euphotic depth, $z_{eu}$. Most recently, a synthetic database was also developed by the first NASA PACE (Plankton, Aerosol, Cloud, ocean Ecosystem) Science Team where the ocean contribution to the top of the atmosphere radiances were simulated by HydroLight (Craig et al., 2020). These simulations were performed from 350 to 800 nm with a 5 nm step for a cloudless sky, three sun zenith angles (10°, 30°, and 60°), wind speed of 5 m s$^{-1}$, and a set of 720 IOP combinations driven by $a_{ph}(\lambda)$. The publicly available output of these HydroLight simulations is $R_{rs}(\lambda)$.

While these existing synthetic databases have offered valuable information to the ocean color radiometry (OCR) community, especially for the purpose of algorithm development where the ocean AOPs are linked to IOPs, there are several reasons that have motivated the present study aiming at generating a new optical synthetic database. First, the inelastic Raman scattering and fluorescence processes were ignored in the previous RT simulations. These inelastic radiative processes are known to be important for simulating realistic characteristics of light fields within and leaving the ocean, including $R_{rs}(\lambda)$ that is a primary optical quantity used in ocean color remote sensing. For example, Raman scattering by water molecules may have an important influence on light within and leaving the ocean and AOPs, especially in the green and red parts of the spectrum (e.g., Marshall and Smith, 1990; Stavn, 1993; Sugihara et al., 1984; Westberry et al., 2013). Second, the three synthetic databases described above are based on the use of the spectral pure seawater absorption, $a_w(\lambda)$, and scattering $b_w(\lambda)$, coefficients values as defined by Pope and Fry (1997) and Morel (1974) in the visible part of the spectrum, respectively. However, more recent measurements and theoretical considerations provide new recommendations for spectral values of $a_w(\lambda)$ and $b_w(\lambda)$ (IOCCG Protocol Series, 2018; Zhang and Hu, 2009; Zhang et al., 2009). Third, the probability distributions of different IOPs that were used as input to previous RT simulations do not appear to match well with the IOP distributions observed in extensive field datasets or satellite-derived datasets

representing the global ocean. This issue may have a biasing effect when the synthetic databases are used to
develop the optical algorithms based on the AOP vs. IOP relationships, especially when the underlying goal is to
represent a broad range of IOPs encountered within the global ocean, even if the primary interest is in open-ocean
pelagic environments. Finally, the previous synthetic databases were developed specifically for OCR-oriented
studies and the publicly accessible data generally include only the surface reflectances, $R_{rs}(\lambda)$, $R(\lambda)$, $r_{rs}(\lambda)$, and
$K_d(\lambda)$ at selected depths. These databases do not include many of the various output variables obtained from RT
simulations, such as the various underwater AOPs, which can be useful in supporting a broader range of studies in
ocean optics beyond ocean color remote sensing.
In this article, we present a new synthetic optical database generated using RT simulations that addresses some
of the limitations of similar databases developed in the past. First, we describe the development of the synthetic
IOP dataset that is required to run RT simulations. The key roles in this development are played by the measured
data of phytoplankton absorption coefficient and desired consistency between the probability distributions of
synthetic IOPs and the global distributions based on satellite observations. Following this, we describe different
configurations of RT simulations that were performed with the HydroLight code. The next section is dedicated to
consistency between the new optical synthetic database and in situ data, including some previously reported
empirical relationships. We provide example illustrations of consistency for both the IOP and AOP data. The
closing section summarizes the structure of synthetic database files and provides example illustration of one output
radiometric variable, the spectral downwelling plane irradiance, calculated with RT simulations.
**2 Development of synthetic dataset of seawater inherent optical properties**
2.1 General overview of methodology
The scope of the synthetic database generated with RT simulations and the degree of its representativeness of
diverse marine optical environments within the global ocean depend most critically on a dataset of seawater IOPs
that are used as input to RT simulations. In the present study, our approach to generate the IOP dataset was driven
largely by an underlying goal to obtain the probability distributions of IOPs that are generally consistent with the
distributions observed in the global ocean dominated by open-ocean pelagic environments. The key IOPs involved
in the creation of our IOP dataset include the spectral absorption and backscattering coefficients associated with
the main categories of seawater constituents representing suspended particulate matter and CDOM. Specifically,
the absorption coefficients of the different constituents are the spectral absorption coefficients of phytoplankton,
$a_{ph}(\lambda)$, non-algal particles, $a_d(\lambda)$, and CDOM, $a_g(\lambda)$. Note that the sum $a_{ph}(\lambda) + a_d(\lambda) = a_p(\lambda)$ represents the
particulate absorption coefficient with combined contributions of phytoplankton and non-algal particles, and the
sum $a_d(\lambda) + a_g(\lambda) = a_{dg}(\lambda)$ represents the non-phytoplankton absorption coefficient with combined contributions
of non-algal particles and CDOM. The backscattering coefficient of the different constituents are the spectral
backscattering coefficients of phytoplankton, $b_{b-ph}(\lambda)$, and non-algal particles, $b_{b-d}(\lambda)$, such that the sum $b_{b-ph}(\lambda) +$
$b_{b-d}(\lambda) = b_{bp}(\lambda)$ is the particulate backscattering coefficient.
Among these constituent IOPs, the phytoplankton absorption coefficient, $a_{ph}(\lambda)$, plays the most fundamental
role in the creation of the synthetic dataset of IOPs in this study. The $a_{ph}(\lambda)$ spectra in this dataset were derived
from actual measurements of phytoplankton absorption made on near-surface samples collected across diverse
oceanic environments. Thus, the $a_{ph}(\lambda)$ data are not "synthetic" in a sense that these data were not obtained from
a modeling approach although some spectral interpolation or extrapolation was applied to measured data as
described in more detail below. In contrast, the remaining four constituent IOPs in the IOP dataset, i.e., $a_d(\lambda)$,
$a_g(\lambda)$, $b_{b\text{-}ph}(\lambda)$, and $b_{b\text{-}d}(\lambda)$, are "synthetic" in a sense that they are entirely based on calculations using a modeling
approach with some assumptions about the magnitude and spectral behavior of the modeled IOPs. Importantly,
the measured values of $a_{ph}(\lambda)$ were used in the calculations of these IOPs. These calculations are also described in
detail below. Thus, each combination of the five constituent IOPs in the synthetic IOP dataset consists of the
measured $a_{ph}(\lambda)$ and the calculated $a_d(\lambda)$, $a_g(\lambda)$, $b_{b\text{-}ph}(\lambda)$, and $b_{b\text{-}d}(\lambda)$ where the results of these calculations depend
on the measured $a_{ph}(\lambda)$. As a result of this approach, it would seem justifiable to refer to the created IOP dataset
as a quasi-synthetic dataset. For simplicity, however, we refer to it as the synthetic IOP dataset while bearing in
mind that $a_{ph}(\lambda)$ spectra were derived from measurements.
2.2 Description of in situ dataset
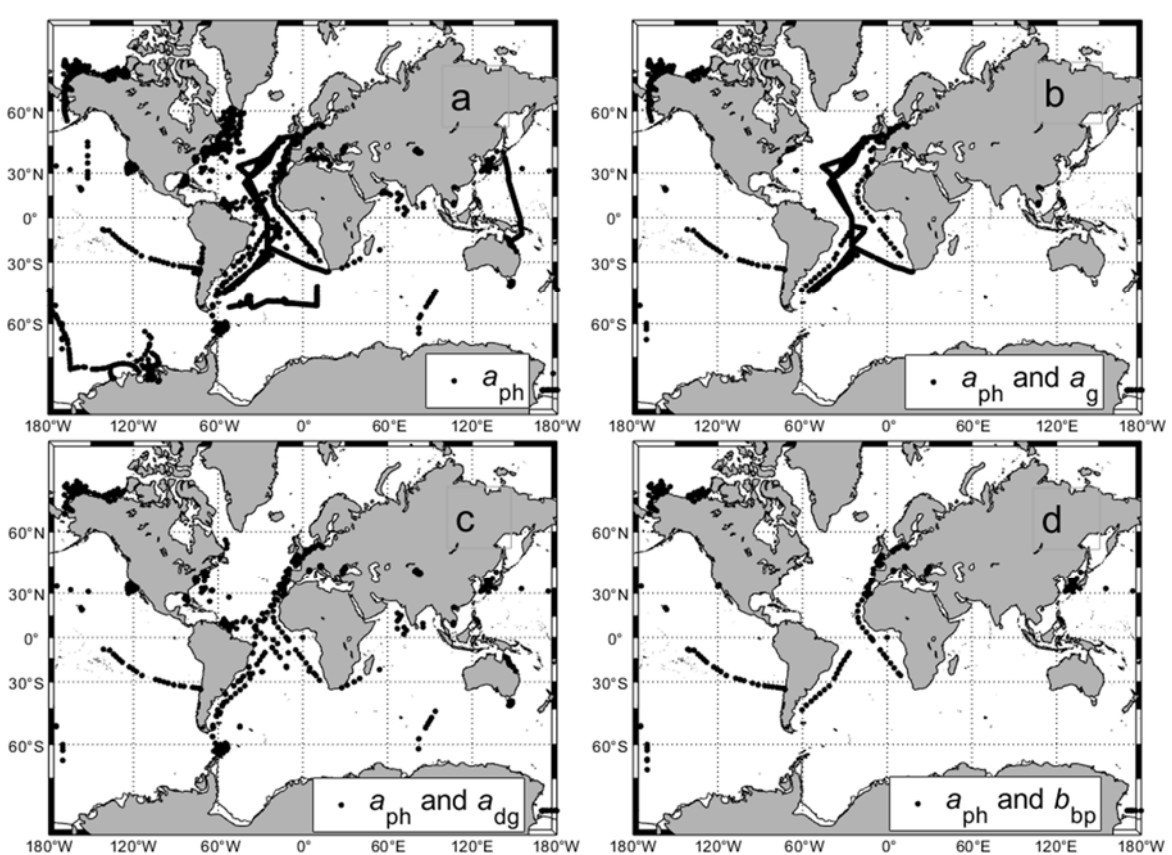

Figure 1. Location of oceanographic stations where in situ measurements were collected for (a) $a_{ph}(\lambda)$, the number of
measurements $N = 4382$; (b) $a_{ph}(\lambda)$ and $a_g(\lambda)$, the number of matchup measurements $N = 2206$; (c) $a_{ph}(\lambda)$ and $a_{dg}(\lambda)$, the
number of matchup measurements $N = 813$; and (d) $a_{ph}(\lambda)$ and $b_{bp}(\lambda)$, the number of matchup measurements $N = 775$.

Figure 1a depicts the location of oceanographic stations where the near-surface measurements of $a_{ph}(\lambda)$ were
made. As shown, these measurements were collected across diverse open ocean and coastal environments and their
total number is 4382 that constitutes the initial field dataset of $a_{ph}(\lambda)$ considered in this study. Figure 1 also shows
the location of stations where coincident measurements are available for the pairs of IOP coefficients, namely
$a_{ph}(\lambda)$ and $a_g(\lambda)$ (Fig. 1b), $a_{ph}(\lambda)$ and $a_{dg}(\lambda)$ (Fig. 1c), and $a_{ph}(\lambda)$ and $b_{bp}(\lambda)$ (Fig. 1d). We recall that while the in
situ data of $a_g(\lambda)$, $a_{dg}(\lambda)$, and $b_{bp}(\lambda)$ were not used in the development of synthetic IOP dataset, they were assembled
for the purpose of comparison with corresponding coefficients that were calculated and included in the synthetic
IOP dataset. Many in situ data of IOP coefficients used in the present study were collected in previous studies
(e.g., Reynolds et al., 2001; Babin et al., 2003; Loisel et al., 2007 ; Claustre et al., 2008; Huot et al., 2008; Stramski
et al., 2008; Lubac et al., 2008 ; Loisel et al., 2009;  Bricaud et al., 2010; Loisel et al., 2011; Antoine et al., 2011;
Neukermans et al., 2012; Uitz et al., 2015; Neukermans et al., 2016; Reynolds et al., 2016; Aurin et al., 2018;
Reynolds and Stramski, 2019; Stramski et al., 2019). Some data are described in publications devoted to
compilation of various datasets (Valente et al., 2019; Casey et al., 2020) and are included in several databases
(e.g., SeaBASS, CoastlOOC, BOUSSOLE, and GOCAD). As the IOP coefficients in the in situ dataset were
measured over a broad range of trophic and environmental conditions, their spectral values span more than 3 or 4
orders of magnitude. This large dynamic range is illustrated in terms of probability distributions at selected light
wavelengths, i.e., at 440 nm for the constituent absorption coefficients and 550 nm for the particulate
backscattering coefficient (Fig. 2).

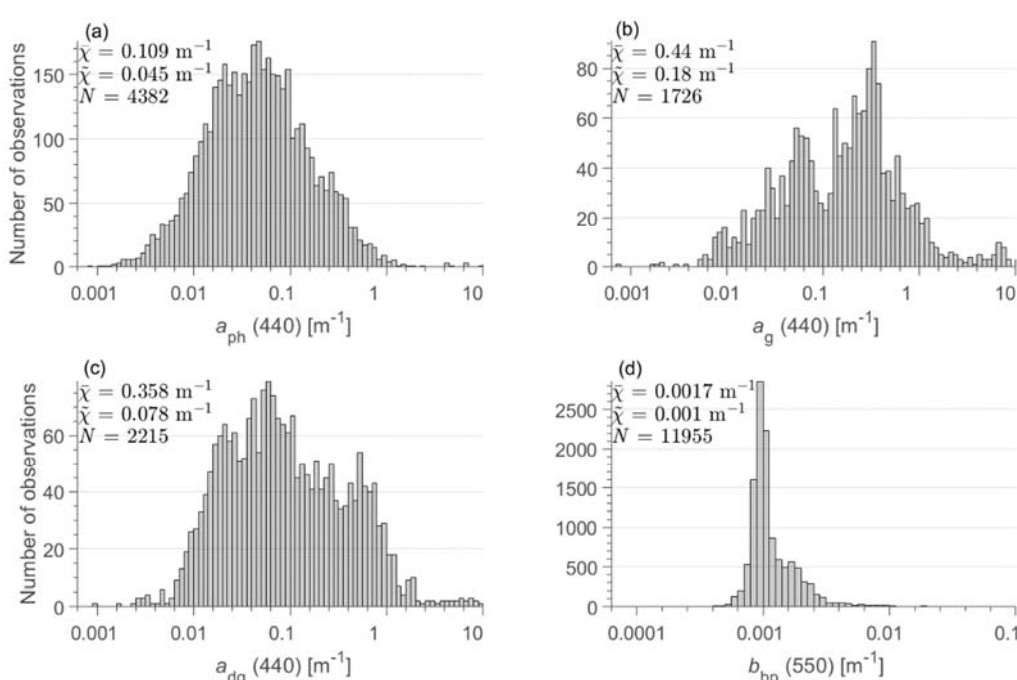

Figure 2. Histograms and relevant statistical parameters of field measurements of (a) $a_{ph}(440)$, (b) $a_g(440)$, (c) $a_{dg}(440)$, and
(d) $b_{bp}(550)$. $N$ is the number of measurements, and $\overline{x}$ and $\tilde{x}$ are the mean and median values of each IOP, respectively.

2.3 Generation of the dataset of hyperspectral $a_{ph}(\lambda)$
The first task necessary for development of the synthetic IOP dataset was to assemble data of hyperspectral
absorption coefficient of phytoplankton, $a_{ph}(\lambda)$, from field measurements collected across diverse open ocean and
coastal environments (Fig. 1a, Fig. 2a). These $a_{ph}(\lambda)$ data were obtained with the filter-pad spectrophotometric
method as a difference between the measurements of $a_p(\lambda)$ and $a_d(\lambda)$ (Kishino et al., 1985; IOCCG Protocol Series,
2018). Historically, most of these measurements were acquired with the transmittance configuration of the filter-
pad method and such measurements are included in our dataset. However, some data in our dataset were obtained
with the inside integrating-sphere configuration of the filter-pad method, which is superior to the transmittance
configuration of measurement (Stramski et al., 2015; IOCCG Protocol Series, 2018).
A significant portion (23.7%) of the initial dataset of $a_{\mathrm{ph}}(\lambda)$ consisting of 4382 measurements covers a spectral
range from 400 to 750 nm with high spectral resolution of data reported at 1 nm interval. In some cases, the original
measurements extended to near-UV spectral region and/or longer wavelengths in the near-infrared spectral region
(800 or 850 nm). The data beyond 750 nm are not used in this study because our RT simulations target the spectral
range from 350 to 750 nm. It is notable that the absorption measurements of marine particles and phytoplankton
are generally unavailable or are not reported in the UV because of increased methodological challenges and
uncertainties in this spectral region (Stramski et al., 2015; IOCCG Protocol Series, 2018; Kostakis et al., 2021).
As a result, only a relatively small fraction of $a_{\mathrm{ph}}(\lambda)$ measurements in the initial dataset were reported in the near-
UV region. In addition, the initial dataset included a relatively large fraction of $a_{\mathrm{ph}}(\lambda)$ measurements that were
reported at wavelength intervals larger than 1 nm. These lower resolution data (hereafter referred to as
multispectral) ranged from a small wavelength interval of 2 nm to data reported at more limited number of
wavelengths (as small as <10) within the visible spectral range. It is likely that the multispectral data available
from some data sources that we used in this study were originally measured at higher spectral resolution but
eventually were reported only for some selected wavelengths, such as those corresponding to spectral bands
available on satellite ocean color sensors.

249        The first objective of the analysis of $a_{\mathrm{ph}}(\lambda)$ was to consider the initial $a_{\mathrm{ph}}(\lambda)$ dataset within the 400–750 nm

range and convert the measurements that were reported at lower spectral resolution to uniformly hyperspectral
data at 1 nm interval. In this analysis, all measurements originally available at 1 nm interval were considered to
provide reference spectral shape functions of $a_{\mathrm{ph}}(\lambda)$. The originally multispectral data of $a_{\mathrm{ph}}(\lambda)$ were converted to
hyperspectral data using several different approaches depending on the spectral features of lower resolution data.
One approach utilized the reference spectral shape functions of $a_{\mathrm{ph}}(\lambda)$ and was applied to multispectral $a_{\mathrm{ph}}(\lambda)$ data
if they were reported at fewer wavelengths than 100. In this case, a given multispectral spectrum of $a_{\mathrm{ph}}(\lambda)$ was
converted to hyperspectral spectrum using a specific hyperspectral measurement that exhibited the highest
correlation with the multispectral measurement under consideration. The correlation coefficient was calculated
using the spectral data available at common wavelengths of considered pair of spectra. A necessary condition to
proceed with a conversion of a given multispectral spectrum to hyperspectral spectrum was a correlation
coefficient of 0.95 or higher. If this condition was satisfied, the multispectral data were converted to hyperspectral
data so that the created hyperspectral spectrum maintained the magnitude of multispectral measurement in the
range of 440–450 nm and had the spectral shape of the reference hyperspectral measurement. An alternative
approach to convert multispectral data to hyperspectral data involved a linear interpolation of multispectral data.
This approach was used when the multispectral data were reported at relatively small wavelength intervals (at least
100 spectral data available between 400 and 750 nm) or when the correlational analysis described above did not
yield the correlation coefficient of 0.95 or higher (5.2% of the multispectral data). The original multispectral
spectra which did not include data below 450 nm or fell into the category of data subject to linear interpolation but
had no data above 700 nm were rejected from further analysis. For all hyperspectral spectra that passed the above-
described analysis and criteria (i.e., 2204 spectra that included both the 593 original hyperspectral measurements
and 1611 hyperspectral spectra created from multispectral data), the null-point correction was applied by
subtracting the average value of $a_{\mathrm{ph}}(\lambda)$ in the 745–750 nm range from all spectral values in the 400–750 nm range.

272        The next step of analysis was to extend all null-point corrected spectra of $a_{\mathrm{ph}}(\lambda)$ that cover the 400–750 nm

range into the UV spectral region. The primary focus was on the 350–400 nm range because our RT simulations
were designed to provide output results in the 350–750 nm range. For this purpose we used a separate subset of
reference hyperspectral measurements of $a_{ph}(\lambda)$ that includes the near-UV spectral region. This reference subset
of data consisted of 233 measurements collected across bio-optically diverse marine environments in the Pacific
and Atlantic Oceans and western Arctic seas. The majority of these 233 spectra (170) were collected with the
inside integrating-sphere configuration of filter-pad method, while the remaining 63 measurements were done
using either the transmittance or transmittance-reflectance filter-pad configuration (Zheng et al., 2014). A
correlational analysis was applied to pairs of spectra, each consisting of a spectrum covering the 400–750 nm range
and a reference spectrum covering the 350–750 nm range. The correlation coefficient was calculated using data at
common wavelengths from the 400–750 nm range. The reference spectrum that yielded the highest correlation
with the investigated 400–750 nm spectrum was selected as a basis for extrapolation of the investigated spectrum
into the 350–400 nm range. This extrapolation ensured that a given investigated spectrum maintained its magnitude
at 400 nm and its extrapolated near-UV portion had the spectral shape of the selected reference spectrum. The final
aspect of extrapolation in the UV is related to the spectral range 300–350 nm. The rationale for IOP data extending
to 300 nm is to ensure that the results of RT simulations that start at 350 nm account for possible effects of Raman
scattering by water molecules in the UV spectral region. Therefore, for the 300–350 nm range we simply assumed
that $a_{ph}(\lambda)$ in this range is equal to $a_{ph}(350)$. The limitation associated with this assumption is not considered to be
serious given the limited role of the 300–350 nm range in the RT simulations and weak Raman scattering effects
in UV spectral region. Example spectra of $a_{ph}(\lambda)$ in the 350–750 nm range from contrasting marine environments
are presented in Fig. 3. These examples show significant variation in both the magnitude and spectral shape of
$a_{ph}(\lambda)$.

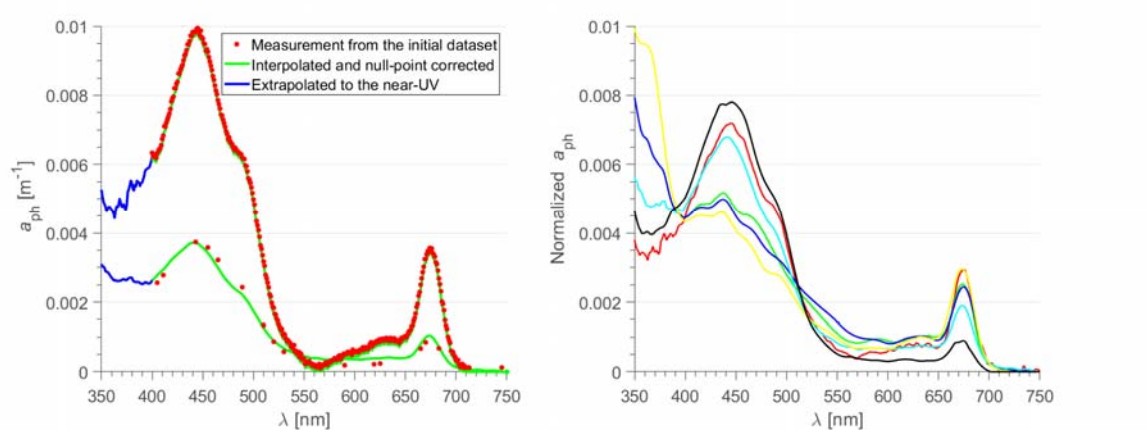

Figure 3. (a) Two example spectra of $a_{ph}(\lambda)$ from contrasting oceanic environments. For each example $a_{ph}(\lambda)$, two spectra are
displayed, namely the measurement from the initial $a_{ph}(\lambda)$ dataset shown at the original wavelength intervals (red points) and
the spectrum after interpolation to 1 nm intervals (if required) and null-point correction (continuous lines). The UV portion of
the latter was obtained by extrapolation based on reference data in the UV (see text for details). (b) Example of normalized
$a_{ph}(\lambda)$ spectra illustrating the variability of the spectral shape of the $a_{ph}(\lambda)$ database. These spectra have been normalized to
their integral.

2.4 Generation of the complete IOP dataset
In the next step of analysis, the subset of 2204 $a_{ph}(\lambda)$ spectra that was created from the initial $a_{ph}(\lambda)$ dataset as
described above was subject to additional modifications to ensure that the final $a_{ph}(\lambda)$ dataset is characterized by

the probability distribution that resembles the distribution representative of the global ocean. This process and background information on the motivation for such adjustments in the probability distribution are described below.

When the end goal is to achieve a high degree of representativeness of global ocean like in this study, the process of assembling in situ datasets of IOPs is unavoidably subject to limitations, even if relatively large amount of data from many field experiments and cruises are considered. This is mainly because the global ocean is dominated by vast areas of open-ocean pelagic environments and the amount of IOP data collected in these environments is disproportionally limited compared to amount of data collected in coastal regions that represent a relatively small portion of the global ocean. Thus, the probability distributions based on in situ datasets, such as those presented in Fig. 2, are expected to deviate from the probability distributions representative of the global ocean. In particular, the maxima of probability distributions and the measures of central tendency, such as the median and mean values, obtained from compilations of relatively large amount of in situ IOP data (such as in Fig. 2) are expected to be shifted to larger values compared to actual global distributions because the IOPs exhibit a general tendency of higher values in coastal regions compared to open ocean environments. While this issue has been recognized, it has not been addressed or resolved in various studies that focus on global ocean color applications. For example, the current global ocean color algorithms for estimating chlorophyll-a concentration (Chla) are based on relatively large amount of in situ data whose probability distribution is shifted significantly to higher Chla compared with the global Chla distribution (O'Reilly and Werdell, 2019). Similarly, in the development of previous synthetic optical databases with RT simulations (e.g., IOCCG Report, 2006), no special attempt was made to ensure consistency between the probability distributions of input IOP data and the distributions expected for global ocean. In the recent development of refined global ocean color algorithms for estimating the concentration of particulate organic carbon (POC), the in situ dataset was assembled with a goal to achieve reasonable consistency with a global POC distribution (Stramski et al., 2022). This goal was, however, achieved at the expense of significant reduction in the amount of accepted in situ data compared to the size of overall pool of available in situ data.

In this study our goal was to create a relatively large synthetic IOP dataset based on the initial dataset of several thousand measurements of spectral $a_{ph}(\lambda)$, so that the probability distributions of IOPs in the final synthetic dataset are reasonably consistent with the expected distributions representative of the global ocean. As described above, the initial field dataset in support of this process consisted of 4382 spectra of $a_{ph}(\lambda)$ and this number was further reduced to 2204 spectra that were accepted as a result of analysis and some criteria applied to the initial dataset. This reduced dataset of accepted $a_{ph}(\lambda)$ spectra was then further modified to ensure that the final probability distribution of $a_{ph}(440)$ resembles the global distribution of $a_{ph}(440)$. The global probability distribution of $a_{ph}(440)$ was estimated using retrievals of $a_{ph}(440)$ from satellite ocean color data. Specifically, we used global satellite observations made with the ocean color sensor OLCI (Ocean and Land Colour Instrument) deployed on the Sentinel-3 mission (Donlon et al., 2012) from the period December 1, 2020 through November 30, 2021. The weekly data product of remote-sensing reflectance $R_{rs}(\lambda)$ at 4 km$^2$ spatial resolution was used as input to the 3-step semi-analytical algorithm (3SAA) to derive $a_{ph}(443)$ as described in Jorge et al (2021). The $a_{dg}(443)$ and $b_{bp}(\lambda)$ coefficients were also derived from this algorithm. In general, the 3SAA first derives the diffuse attenuation coefficient for downwelling plane irradiance averaged within the surface layer down to the first attenuation depth, $<K_d(\lambda)>_1$, from $R_{rs}(\lambda)$, and then utilizes the inverse model LS2 (Loisel et al., 2018) to derive the total absorption, $a(\lambda)$, and backscattering, $b_b(\lambda)$, coefficients from $R_{rs}(\lambda)$ and $<K_d(\lambda)>_1$. After subtracting the pure

seawater contributions, the non-water absorption, $a_{\text{nw}}(\lambda)$, and the particulate backscattering, $b_{\text{bp}}(\lambda)$, coefficients are obtained. Finally, $a_{\text{ph}}(\lambda)$ and $a_{\text{dg}}(\lambda)$ are derived from $a_{\text{nw}}(\lambda)$ using an optimization algorithm of Zhang et al. (2015) with modifications that account for differences in optical water types defined in terms of different spectral shapes of $R_{\text{rs}}(\lambda)$ (Mélin and Vantrepotte, 2015). While the original classification of Mélin and Vantrepotte (2015) includes 16 optical water classes (OWC), the derivation of $a_{\text{ph}}(\lambda)$ and $a_{\text{dg}}(\lambda)$ from the 3SAA additionally included a 17th OWC to improve the representation of ultraoligotrophic waters such as those found in the South Pacific Gyre (Morel et al., 2007; Claustre et al., 2008; Stramski et al., 2008) and in some areas of the Mediterranean Sea in summer (Loisel et al., 2011). This 17th OWC is described in Jorge et al. (2021).

The 3SAA does not yield the separate contributions of CDOM, $a_{\text{g}}(\lambda)$, and non-algal particles, $a_{\text{d}}(\lambda)$, to the overall non-phytoplankton absorption coefficient, $a_{\text{dg}}(\lambda)$. Therefore, we also used another semi-analytical model (CDOM-KD2) described in Bonelli et al. (2021) to estimate $a_{\text{g}}(443)$ from OLCI-derived $R_{\text{rs}}(\lambda)$. Having $a_{\text{dg}}(\lambda)$ from the 3SAA and $a_{\text{g}}(\lambda)$ from the CDOM-KD2, the non-algal particulate absorption, $a_{\text{d}}(\lambda)$, was obtained as a difference $a_{\text{dg}}(\lambda) - a_{\text{g}}(\lambda)$. As a result of this analysis, we obtained a dataset of satellite-derived constituent absorption coefficients, $a_{\text{ph}}(443)$, $a_{\text{g}}(443)$, $a_{\text{d}}(443)$, and $a_{\text{dg}}(443)$, as well as the particulate backscattering coefficient, $b_{\text{bp}}(550)$, where we focused on the spectral band near 440 nm for absorption and 550 nm for backscattering.

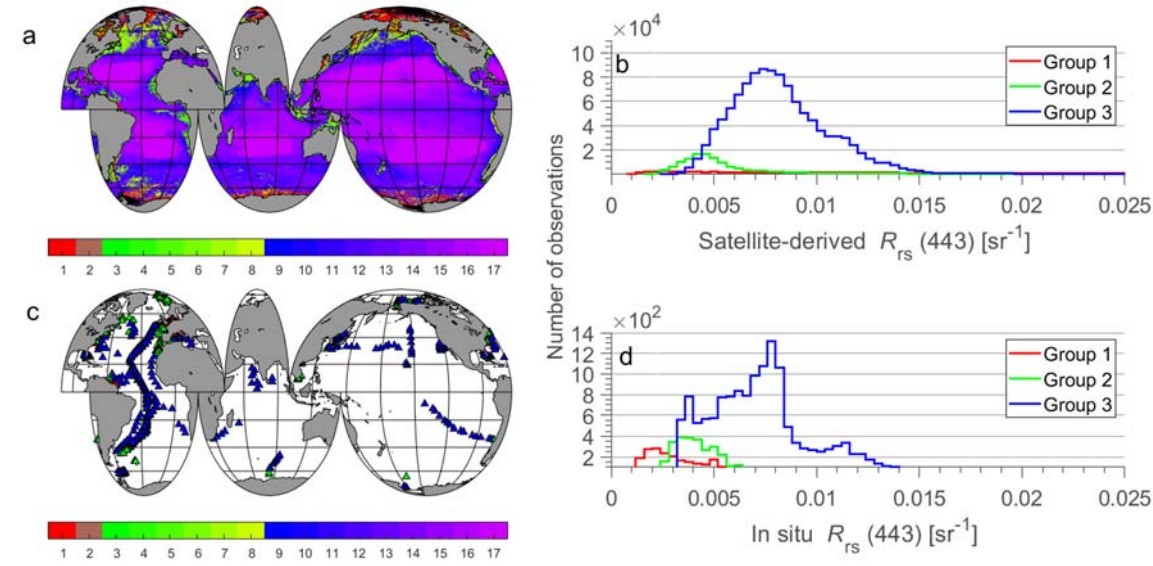

Figure 4. (a) Global map illustrating the distribution of seventeen optical water classes estimated from monthly $R_{\text{rs}}(\lambda)$ values derived from satellite observations with ocean color sensor OLCI from December 2020 through November 2021 (weekly products at 4km$^2$). The color bar scale refers to optical water classes. (b) Histogram of OLCI-derived $R_{\text{rs}}(443)$ for the three optical water groups (see text for details). (c) Location of oceanographic stations where in situ measurements of $R_{\text{rs}}(\lambda)$ were collected and used to analyze the consistency of the synthetic dataset with field measurements. (d) Histograms of in situ measurements of $R_{\text{rs}}(443)$ for the three optical water groups.

For illustrative purposes Fig. 4a depicts the spatial distribution of 17 optical water classes (OWCs) over the global ocean obtained from satellite OLCI data following the methodology of Mélin and Vantrepotte (2015). For further illustrative purposes, these 17 OWCs were grouped into 3 optical water groups (OWGs). Group 1 consists of OWC1 and OWC2 which are characterized by high water turbidity such as in coastal areas affected by discharge from large rivers. Although the focus of this study is to create the synthetic datasets representative primarily of

open ocean and moderately turbid coastal waters, an explicit identification of Group 1 data that represent very
turbid waters is of interest for comparisons with the database developed specifically for coastal waters by Nechad
et al. (2015). The second OWG, Group 2, includes 6 OWCs from OWC3 through OWC8. This group represents
mainly productive waters in both coastal and open ocean environments, such as those encountered in the North
Atlantic during a period of phytoplankton bloom (Levy et al., 2005). Finally, Group 3 included the remaining 9
OWCs from OWC9 through OWC17. These water types are observed mainly in mesotrophic and oligotrophic
regions of the global ocean. Based on this classification, 79.6% of OLCI water pixels in Fig. 4a belong to Group
3, 10.8% to Group 2, and 9.6% to Group 1. The histograms of OLCI-derived $R_{rs}(443)$ associated with these three
groups of data are shown in Fig. 4b. For comparative purposes we also assembled a dataset of in situ measurements
of $R_{rs}(\lambda)$, which were collected at various locations within the global ocean (Fig. 4c). The histograms of in situ
$R_{rs}(443)$ associated with Groups 1, 2, and 3 are depicted in Fig. 4d, which show a similar pattern to that in Fig. 4b.
For in situ dataset of $R_{rs}(\lambda)$, 69.2% of data belong to Group 3, 15.7% to Group 2, and 15.1% to Group 1.

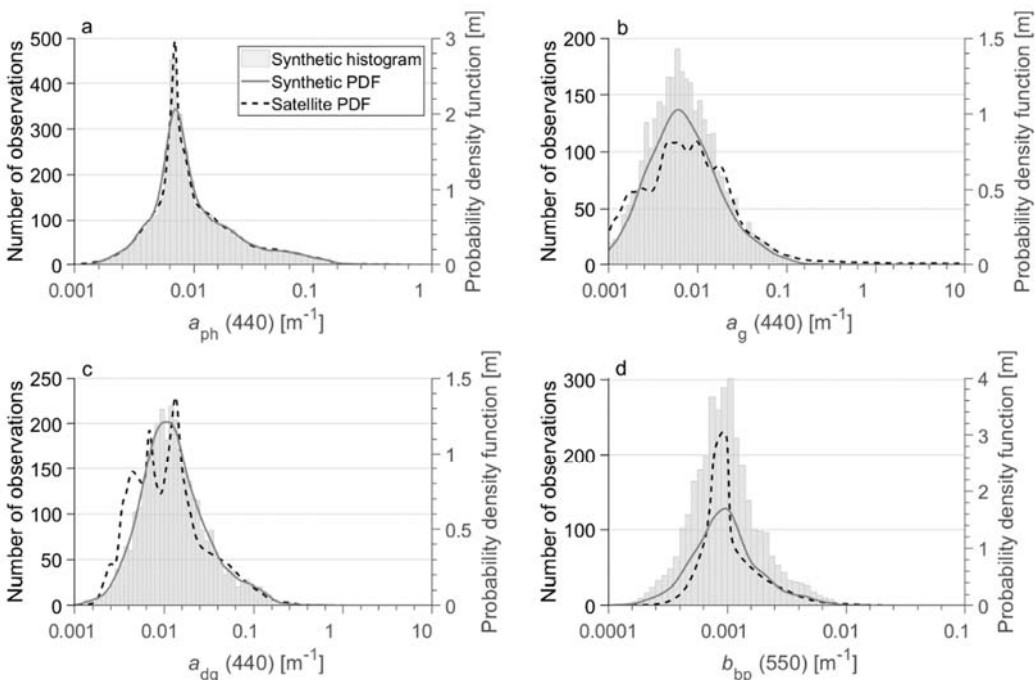

Figure 5. Histograms showing the distribution of the synthetic IOP data used in the present study. The synthetic and satellite-
derived probability density functions (PDFs) for each IOP are represented by the solid and dashed curves, respectively.

The probability density function (PDF) of global satellite-derived $a_{ph}(440)$, $a_g(440)$, $a_{dg}(440)$, and $b_{bp}(550)$ are
depicted in Fig. 5. We note that we refer here to satellite-derived absorption coefficients at 440 nm although they
were derived from OLCI reflectances at 443 nm, which is a minor difference that is inconsequential for the purpose
of this study. The comparison of Fig. 2a and Fig. 5a indicates that the distribution of measured $a_{ph}(440)$ from our
initial field dataset (Fig. 2a) is shifted towards higher values compared to the global distribution of satellite-derived
$a_{ph}(440)$ (Fig. 5a). The probability distribution of reduced dataset of measured $a_{ph}(440)$ ($N = 2204$) that was created
from the initial field dataset of $a_{ph}(\lambda)$ show similar deviations from the global distribution (not shown). Thus, to
create the final dataset of $a_{ph}(\lambda)$ that has the probability distribution of $a_{ph}(440)$ consistent with the global satellite-
derived distribution, we adjusted the number of $a_{ph}(440)$ measurements in each bin of the histogram of the reduced

dataset either by removing the measurements from any given bin or adding the measurements to this bin. The removal or addition of $a_{ph}(440)$ measurements associated with any given bin was done by subjecting all $a_{ph}(440)$ measurements originally contained within a given bin to random selection. Specifically, in the case of addition the randomly selected $a_{ph}(440)$ was added as a replicate of $a_{ph}(440)$ to a given bin. In the case of removal, the randomly selected $a_{ph}(440)$ was simply removed from a given bin. As a result of this process we obtained a modified distribution of measured $a_{ph}(440)$ that is fairly consistent with the satellite-derived distribution of $a_{ph}(440)$. Both the modified histogram and the corresponding modified PDF of measured $a_{ph}(440)$ are depicted in Fig. 5a for comparison with the global satellite-derived distribution. In total, this modified distribution consists of 3320 measurements of $a_{ph}(440)$ and, obviously, each of these measurements at 440 nm has an associated full spectrum of $a_{ph}(\lambda)$ values between 300 and 750 nm. These 3320 spectra of $a_{ph}(\lambda)$ represent one IOP component of the final synthetic IOP dataset.

The full synthetic IOP dataset created in this study consists of 3320 combinations of measured $a_{ph}(\lambda)$ and synthetically-generated $a_d(\lambda)$, $a_g(\lambda)$, $b_{b-ph}(\lambda)$, and $b_{b-d}(\lambda)$. Below is a description of calculations of $a_g(\lambda)$, $a_d(\lambda)$, $b_{b-ph}(\lambda)$, and $b_{b-d}(\lambda)$. We note that all IOP coefficients are expressed in units of [m$^{-1}$] and the light wavelength is in units of [nm].

The four IOP coefficients were calculated using a similar methodology to that applied in previous studies aiming at generation of synthetic ocean optical databases (IOCCG Report, 2006; Craig et al., 2020). Specifically, we used the measured values of $a_{ph}(440)$ as the main driver of calculations of $a_g(\lambda)$, $a_d(\lambda)$, $b_{b-ph}(\lambda)$, and $b_{b-d}(\lambda)$. Thus, the variability in the measured $a_{ph}(440)$, as depicted by the probability distribution of measured $a_{ph}(440)$ in Fig. 5a, is the main source of variability in these four co-existing IOP coefficients. It is notable that the replicate values of $a_{ph}(440)$ present within any given bin of the $a_{ph}(440)$ distribution result in the generation of different values of the four IOP coefficients because the formulas involved in these calculations contain random parameters. The coupling between $a_{ph}(440)$ and CDOM absorption coefficient was defined as:

$$a_g(440) = 10^{(P_1 + \gamma)} \tag{1}$$

where $P_1$ is a parameter related to $a_{ph}(440)$ and $\gamma$ is randomly selected from a predetermined range of values (Table 1). The spectral values of $a_g(\lambda)$ are subsequently determined from:

$$a_g(\lambda) = a_g(440) \, e^{-S_g(\lambda - 440)} \tag{2}$$

where the spectral slope parameter, $S_g$ in units of [nm$^{-1}$], is randomly selected from a predetermined range of values (Table 1). The absorption coefficient of non-algal particles was modeled in a similar fashion:

$$a_d(440) = P_2 \, a_{ph}(440) \tag{3}$$

$$a_d(\lambda) = a_d(440) \, e^{-S_d(\lambda - 440)} \tag{4}$$

where $P_2$ is a parameter related to $a_{ph}(440)$ and the spectral slope parameter $S_d$ [nm$^{-1}$] is randomly selected from a predetermined range of values (Table 1). The parameterizations of $P_1$ and $P_2$ were chosen to match relationships observed with the in situ dataset assembled in this study.

The particulate backscattering is not modeled in terms of the single coefficient, $b_{bp}(\lambda)$, but instead as separate contributions by phytoplankton, $b_{b-ph}(\lambda)$, and non-algal particles, $b_{b-d}(\lambda)$, so that their sum yields $b_{bp}(\lambda)$. In order to calculate $b_{b-ph}(\lambda)$, first the formula that couples $a_{ph}(440)$ with the beam attenuation coefficient of phytoplankton at 550 nm, $c_{ph}(550)$, is used:

$$c_{ph}(550) = P_3 \, \text{Chla}^{0.57} = P_3 \left[ \frac{a_{ph}(440)}{0.05582} \right]^{0.57} \tag{5}$$

where Chla is the concentration of chlorophyll-a in units of [mg m$^{-3}$], 0.05582 [m$^2$ mg$^{-1}$] is the value of chlorophyll-
specific absorption coefficient of phytoplankton at 440 nm, $a_{ph}^*(440)$ (Maritorena et al., 2002), and $P_3$ is a
parameter with a randomly selected value from a predetermined range (Table 1). The exponent value of 0.57 is
based on the study of Voss (1992). Subsequently, the spectral values of phytoplankton beam attenuation coefficient
are calculated from:
$$c_{ph}(\lambda) = c_{ph}(550) \left(\frac{550}{\lambda}\right)^{S_{c-ph}} \tag{6}$$
where the spectral slope parameter, $S_{c-ph}$ [dimensionless], is calculated using both $a_{ph}(440)$ and a random number
generator (Table 1). Next, the spectral scattering coefficient of phytoplankton is determined:
$$b_{ph}(\lambda) = c_{ph}(\lambda) - a_{ph}(\lambda) \tag{7}$$
where the spectral values of $a_{ph}(\lambda)$ are from the same measured spectrum as the value of $a_{ph}(440)$ in Eq. (5). Finally,
the spectral backscattering coefficient of phytoplankton is calculated from:
$$b_{b-ph}(\lambda) = 0.01\, b_{ph}(\lambda) \tag{8}$$
where 0.01 is the value of backscattering ratio of phytoplankton, $\tilde{b}_{b-ph}$, assumed to be constant and independent
of light wavelength (IOCCG, 2006; Loisel et al., 2007; Whitmire et al., 2010). We note that $b_{b-ph}(\lambda)$ is not
required as input to our radiative transfer simulations but $b_{ph}(\lambda)$ is needed.
To calculate the backscattering coefficient of non-algal particles, $b_{b-d}(\lambda)$, the phytoplankton absorption at 440
nm is first coupled with the scattering coefficient of non-algal particles at 550 nm, $b_d(550)$, using the following
relationship:
$$b_d(550) = P_4\, \text{Chla}^{0.766} = P_4 \left[\frac{a_{ph}(440)}{0.05582}\right]^{0.766} \tag{9}$$
where the parameter $P_4$ is randomly selected from a predetermined range (Table 1) and the value of 0.05582 is
$a_{ph}^*(440)$ as explained in relation to Eq. (5). The exponent value of 0.766 is based on the study of Loisel et al.
(1998). Then, the spectral values of non-algal scattering coefficient are calculated from:
$$b_d(\lambda) = b_d(550) \left(\frac{550}{\lambda}\right)^{S_{b-d}} \tag{10}$$
where the spectral slope parameter, $S_{b-d}$ [dimensionless], is calculated using both $a_{ph}(440)$ and a random number
generator (Table 1). In the final step, the spectral backscattering coefficient of non-algal particles is calculated as:
$$b_{b-d}(\lambda) = 0.018\, b_d(\lambda) \tag{11}$$
where the constant 0.018 is the backscattering ratio of non-algal particles, $\tilde{b}_{b-d}$. This value was proposed by
Mobley (1994) and was derived by averaging three particle phase functions measured in oceanic waters by
Petzold (1972). Again, we note that $b_{b-d}(\lambda)$ is not required as input to radiative transfer simulations but $b_d(\lambda)$ is
needed. The spectral slope of $b_{bp}(\lambda)$, $\gamma$, where $b_{bp}(\lambda)$ is obtained as the sum of $b_{b-ph}(\lambda)$ and $b_{b-d}(\lambda)$, has a mean and
standard deviation of $1.10 \pm 0.34$, and exhibits a decreasing trendfrom oligotrophic (where $\gamma$ is around -2) to
eutrophic waters (where the $b_{bp}(\lambda)$ spectrum is nearly flat). These results are in good agreement with previous
studies (Morel and Maritorena, 2001; Loisel et al., 2006; Antoine et al., 2011).



Table 1: Symbols of variables, mathematical expressions, and corresponding equations in the text of the paper. $rng(0,1)$ is a random number between 0 and 1.

| Symbols | Mathematical expression | Equation | Reference |
|---|---|---|---|
| $P_1$ | $0.79 \log_{10}[a_{ph}(440)] - 0.37$ | (1) | This study |
| $\gamma$ | $-0.2 + 0.3 \, rng(0,1)$ | (1) | This study |
| $S_g$ | $(0.02 - 0.01) \, rng(0,1) + 0.01$ | (2) | IOCCG (2006) |
| $P_2$ | $(0.1 + rng(0,0.9)) a_{ph}(440)$ | (3) | This study |
| $S_d$ | $(0.015 - 0.007) \, rng(0,1) + 0.007$ | (4) | IOCCG (2006) |
| $P_3$ | $(0.3 - 0.03) \, rng(0,1) + 0.03$ | (5) | Based on IOCCG (2006) |
| $S_{c-ph}$ | $-0.4 + \dfrac{1.6 + 1.2 \, rng(0,1)}{1 + [\frac{a_{ph}(440)}{0.05582}]^{0.5}}$ | (6) | IOCCG (2006) |
| $P_4$ | $(0.16668 - 0.016668) \, rng(0,1) + 0.016668$ | (9) | Based on IOCCG (2006) |
| $S_{b-d}$ | $-0.5 + \dfrac{2 + 1.2 \, rng(0,1)}{1 + [\frac{a_{ph}(440)}{0.05582}]^{0.5}}$ | (10) | IOCCG (2006) |

The variability of measured $a_{ph}(440)$ illustrated in Fig. 5a along with the dynamic range of parameters $P_1$, $P_2$, $P_3$, $P_4$, the spectral slopes $S_g$, $S_d$, $S_{c-ph}$, and $S_{b-d}$, and the degree of randomness in the selection of these parameters for any given value of $a_{ph}(440)$ that initiates the process of calculating $a_g(\lambda)$, $a_d(\lambda)$, $b_{b-ph}(\lambda)$, and $b_{b-d}(\lambda)$, resulted in the generation of synthetic dataset of these IOP coefficients that cover a wide dynamic range consistent with in situ and satellite observations over the global ocean. Figure 5b,c,d compares the probability distributions of satellite-derived $a_g(440)$, $a_{dg}(440)$, and $b_{bp}(550)$ with the distribution of these coefficients from the final synthetic IOP dataset. This comparison supports the general consistency of the distributions of these IOP coefficients, which is in line with the desired consistency achieved for $a_{ph}(440)$ (Fig. 5a) as discussed earlier in this section. It is also noteworthy that in contrast to this newly created synthetic IOP dataset, the previous synthetic datasets exhibit significant differences between the probability distributions of synthetic IOPs and global distributions based on satellite observations (Fig. 6).

Overall, the above-described synthetic IOP dataset includes 3320 scenarios of non-water IOPs, i.e., IOPs associated with variable contributions of phytoplankton, non-algal particles, and CDOM to optical properties of seawater. In addition to the non-water absorption coefficients, $a_{ph}(\lambda)$, $a_d(\lambda)$, and $a_g(\lambda)$, as well as the non-water scattering coefficients, $b_{ph}(\lambda)$ and $b_d(\lambda)$, the radiative transfer simulations required input of scattering phase functions of particles, specifically for phytoplankton and non-algal particles. We assumed the particulate phase functions proposed by Fournier-Forand (1994) with the backscattering ratio $\tilde{b}_{b-ph} = 0.01$ for phytoplankton and $\tilde{b}_{b-d} = 0.018$ for non-algal particles. Note that while the backscattering ratios are assumed to be spectrally constant, the phase functions vary with light wavelength because of spectral variations of $b_{ph}(\lambda)$ and $b_d(\lambda)$. All IOP data in the final synthetic IOP dataset cover the spectral range from 300 to 750 nm with a 5 nm interval. This wavelength interval is consistent with the intended output of our radiative transfer simulations.

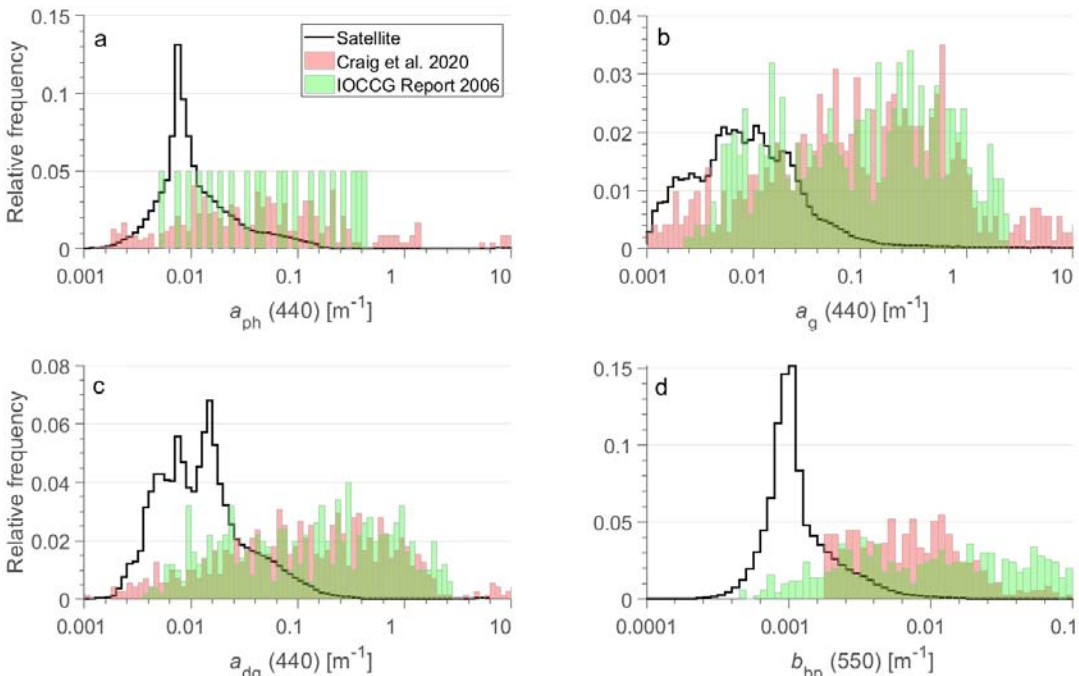

Figure 6. Histograms showing the distribution of IOPs from the synthetic datasets of the IOCCG Report (2006) and Craig et al. (2020) in the green and pink, respectively. The IOP distributions estimated from satellite ocean color observations with OLCI sensor over the global ocean are represented by the black line.

The radiative transfer simulations also required input of the absorption and scattering properties of pure seawater. For the spectral absorption coefficient of pure seawater, $a_w(\lambda)$, we used the values recommended in IOCCG Protocol Series (2018). This recommendation includes the values from Jonasz and Fournier (2007) in the spectral range 300-330 nm, Morel et al. (2007) in the 340-415 nm range, Pope and Fry (1997) in the 420-725 nm range, and Kou et al. (1993) in the 730-750 nm range. The spectral volume scattering function of pure seawater (from which the spectral scattering coefficient and scattering phase function can be obtained) was calculated following Zhang et al. (2009) assuming water temperature of 18°C and salinity of 35‰. The temperature of 18°C is consistent with the mean sea surface temperature (SST) calculated from the monthly global NOAAv2 SST database at 1° spatial resolution from December 1991 through November 2021 (Jérôme Vialard, personal communication, https://www.psl.noaa.gov/data/gridded/data.noaa.oisst.v2.html). The salinity of 35‰ is also consistent with the global surface average (Durack et al., 2013).

## 3  Radiative transfer simulations

The IOP dataset described in section 2, which includes 3320 combinations of non-water IOPs, provided the key input to radiative transfer (RT) simulations that were performed with the HydroLight v5.0 radiative transfer code (Mobley and Sundman, 2008). All RT simulations were run assuming vertically homogeneous IOPs within the water column and infinitely deep ocean, i.e., no effect of seafloor on light field within the water column. For all simulations the computed radiometric and AOP variables were saved into the output data files at 10 cm depth intervals between the ocean surface and the 1 m depth, and at 1 m intervals between the 1 m and 50 m depth. Thus, the primary focus of our RT simulations is on the ocean surface layer that can potentially contribute to light leaving the ocean with significance to remote sensing with spaceborne or airborne optical instruments. All simulations were carried out in the spectral range from 300 to 750 nm using 5-nm spectral bands and the results were produced

for the nominal wavelengths of each of the 81 bands, that is at 350, 355, 360, etc…, 745, 750 nm. The results in the 300–350 nm range were not retained in the output files (that include seawater IOPs, radiometric quantities and AOPs) because this spectral region was included primarily to account for potential effects of inelastic processes at wavelengths longer than 350 nm and, additionally, it is known the uncertainties in the characterization of seawater IOPs can increase significantly at wavelengths shorter than 350 nm.

For 3320 scenarios of input IOPs we performed several separate sets of RT simulations that differed in terms of assumed sea-surface boundary conditions and the inclusion or exclusion of inelastic radiative processes within the water column. The assumptions regarding the sea-surface boundary conditions were the same as in the previous RT simulations described in Loisel et al. (2018). Specifically, all simulations were made under the same assumption of wind speed of 5 m s$^{-1}$, which determines the sea-surface roughness involved in the calculations of transmission and reflection of light at the air-water interface. In all simulations the sky conditions were also assumed to be the same, i.e., clear skies and standard atmosphere. However, three distinct sets of simulations were made for the three values of sun zenith angle, 0°, 30°, and 60°. With regard to consideration of inelastic processes, we also performed three distinct sets of simulations. The first of these sets assumed the absence of inelastic processes in water, that is no Raman scattering by water molecules, no fluorescence by chlorophyll-a, and no fluorescence by CDOM. The second set of these simulations included Raman scattering by water molecules. Finally, the third set included both Raman scattering and chlorophyll-a fluorescence, and this scenario of inelastic processes is expected to generally provide the most realistic simulations of radiative transfer in the ocean surface layer. We note, however, that fluorescence by CDOM was not included in any simulations. The Raman scattering coefficient, phase function, and wavelength distribution function were set to their default values described in HydroLight technical documentation (Mobley, 2012). The quantum efficiency of chlorophyll-a fluorescence, which may exhibit significant variability (nearly 5-fold between about 0.01 and 0.05) in ocean waters (Maritorena et al., 2000; Morrison et al., 2003), was also set to its default value of 0.02 in the HydroLight code. For each scenario of sun zenith angle and inelastic processes, we performed 3320 RT simulations, each for a different combination of seawater IOPs. Thus, given the three sun zenith angles, the three scenarios of inelastic processes, and 3320 combinations of IOPs, overall we performed 29880 simulations. The combination of the synthetic IOP dataset used as input to RT simulations (section 2) and the results for the radiance, other radiometric quantities, and AOPs obtained from these 29880 simulations (described in this section) constitute the synthetic ocean optical database developed in this study.

## 4 Comparisons of the synthetic database with in situ data

In this section we compare the selected spectral IOP coefficients from the synthetic IOP dataset with in situ data of IOPs and the selected spectral AOPs from the synthetic database generated with the RT simulations with in situ data of AOPs. In these comparisons, we also include some empirical relationships between the IOPs or AOPs that were established in previous studies based on the analysis of in situ data.

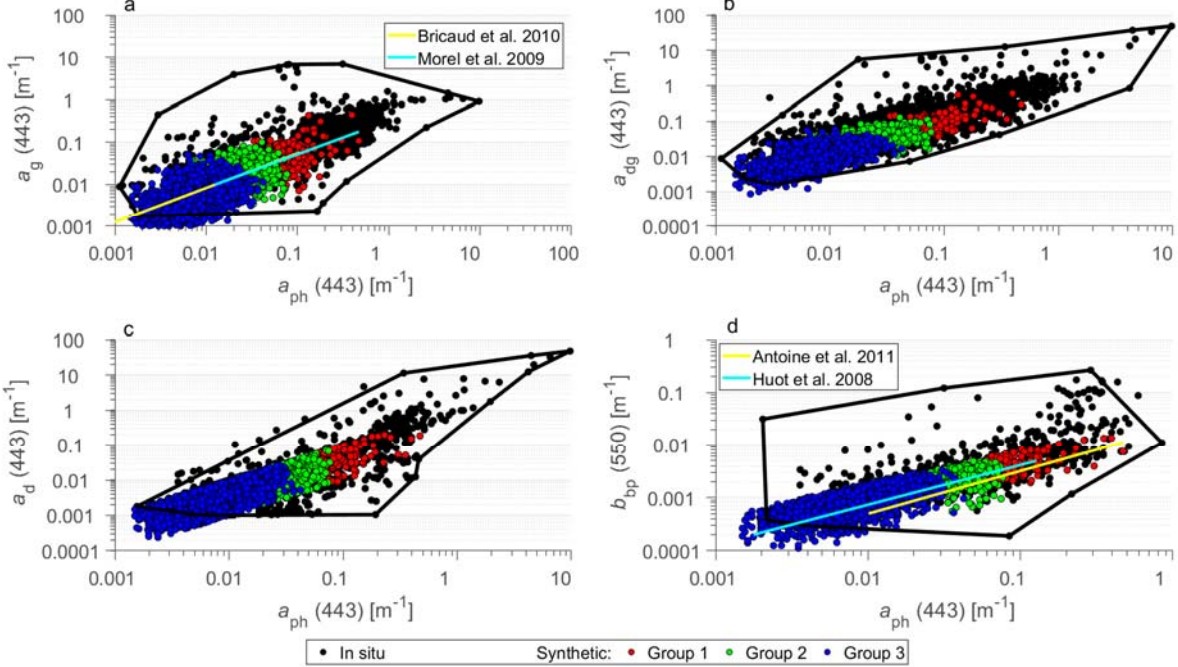

Figure 7. (a) $a_g$(443), (b) $a_{dg}$(443), (c) $a_d$(443), and (d) $b_{bp}$(550) as a function of $a_{ph}$(443) for the in situ dataset (black data
points) and the synthetic dataset (colored data points). The black polygon lines in each panel delimit approximately the scatter
of the in situ data points (black dots). Each color refers to the optical water group as indicated (139, 262, and 2919 data points
for Group 1, 2, and 3, respectively). Empirical relationships previously developed for (a) $a_g$(443) vs. $a_{ph}$(443) and (d) $b_{bp}$(550)
vs. $a_{ph}$(443) are also displayed for comparison. The original relationships were formulated as a function of Chla and the
presented relationships were obtained by converting Chla to $a_{ph}$(443) using the chlorophyll-specific phytoplankton absorption
at 443 nm from Bricaud et al. (1998).

Figure 7 depicts the scatter plots of IOP coefficients, specifically $a_g$(440) vs. $a_{ph}$(440) (Fig. 7a), $a_{dg}$(440) vs.
$a_{ph}$(440) (Fig. 7b), $a_d$(440) vs. $a_{ph}$(440) (Fig. 7c), and $b_{bp}$(550) vs. $a_{ph}$(440) (Fig. 7d). The scatter plots include two
datasets, the in situ dataset and the synthetic dataset as described in section 2. We recall that in both types of
datasets, $a_{ph}$(440) plotted on the *x*-axis is the same because the phytoplankton absorption data used in this study
were obtained from field measurements with no modeling involved. The scatter plots show a significant degree of
overlap which indicates general consistency between the synthetic and in situ datasets. Similar patterns are
observed when the $a_g$(440)/$a_{ph}$(440), $a_{dg}$(440)/$a_{ph}$(440), $a_d$(440)/$a_{ph}$(440), and $b_{bp}$(550)/$a_{ph}$(440) ratios are plotted
as a function of $a_{ph}$(440) (not shown). For illustrative purposes, the data from the synthetic IOP dataset are color
coded to indicate the partitioning of data into the three OWGs, i.e., Groups 1, 2, and 3 that were defined using the
synthetic spectra of $R_{rs}(\lambda)$ generated through RT simulations with input of the synthetic IOP data. As expected,
the data with generally lowest values of IOPs belong to Group 3, the data with intermediate values of IOPs to
Group 2, and the data with the highest IOPs (most turbid waters) to Group 1. We also note the in situ dataset
exhibits somewhat wider dynamic range of variability than the synthetic dataset, especially when the IOP ratios,
$a_g$(440)/$a_{ph}$(440), $a_{dg}$(440)/$a_{ph}$(440), $a_d$(440)/$a_{ph}$(440), and $b_{bp}$(550)/$a_{ph}$(440), are relatively high. While this result
can reflect some degree of intrinsic difference in the dynamic range covered by the two datasets, it must also be
recognized that some variability in the in situ dataset may be associated with the fact that these data were collected
on numerous cruises by different groups of investigators using the methodology (instrumentation, data processing,
data quality control, etc.) that unavoidably was not the same across all different data sources.
For additional comparative purposes, Fig. 7a,d includes a few empirical relationships between the IOPs in
question, which were established in previous studies based on considerable amount of field measurements
collected mostly in open ocean environments. As seen, the relationships between $a_g(440)$ and $a_{ph}(440)$ based on
the studies of Morel (2009) and Bricaud et al. (2010) agree quite well with the central tendency of variation within
our synthetic dataset. We note that Morel (2009) and Bricaud et al. (2010) reported on the relationships between
$a_g(440)$ and Chla that were very similar in these studies. For the purpose of illustration in our Fig. 7a, we replaced
Chla with $a_{ph}(440)$ using the formula Chla = $a_{ph}(440)/0.05582$. Similarly, the studies of Huot et al. (2008) and
Antoine et al. (2011) reported on empirical relationships between $b_{bp}(\lambda)$ and Chla. After converting Chla to
$a_{ph}(440)$ as mentioned above, these two relationship are plotted in Fig. 7d. Although these two relationships have
different slopes, they are both generally consistent with the average trend of variation in the synthetic dataset.
The radiative transfer is driven mainly by two ratios of IOPs which are the scattering to absorption ratio,
$b(\lambda)/a(\lambda)$, which controls the number of scattering events (Morel and Gentili, 1991), and the molecular to total
scattering ratio, $b_w(\lambda)/b(\lambda)$, which is the parameter controlling the weighted sum of the particle scattering and
molecular scattering phase functions (Morel and Loisel, 1998; Loisel and Stramski, 2000). Figure 8 shows the
distribution of these two ratios at 440 nm for the synthetic dataset. The $b_w(440)/b(440)$ and $b(440)/a(440)$ ratios
range between about 0 and 0.2 and 0.5 and 10, respectively, which is consistent with previous models developed
for Case-1 waters (Figs. 2 and 3 in Morel and Gentili, 1991; Fig. 2 in Morel and Loisel, 1998; ).

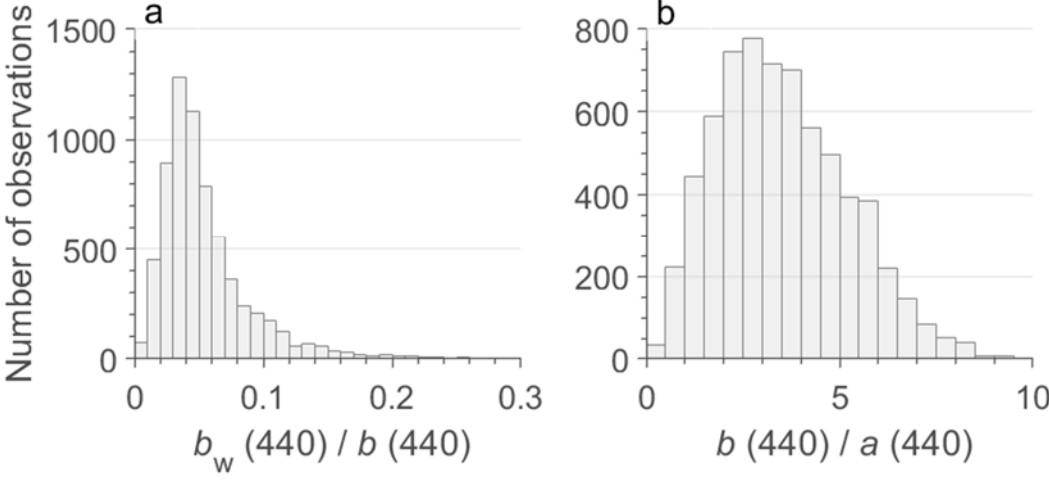


Figure 8. (a) Histograms of (a) $b_w(440)/b(440)$ and (b) $b(440)/a(440)$ for the synthetic dataset.

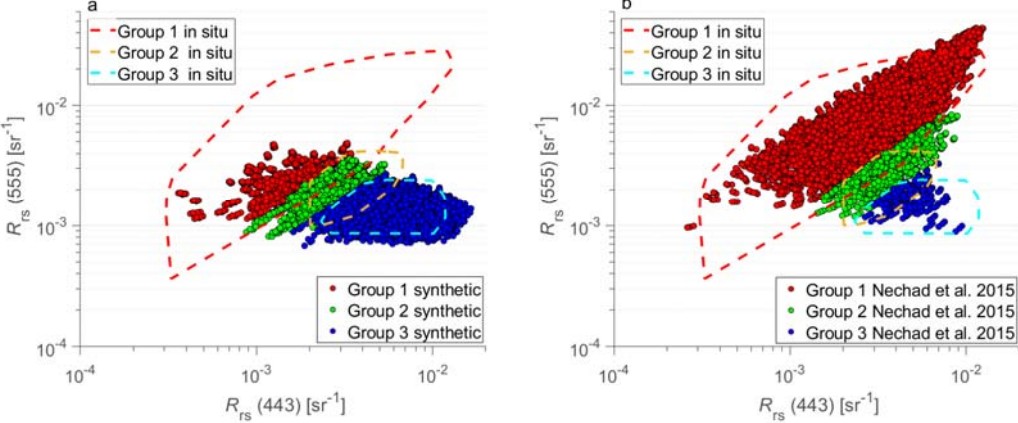

Figure 9. (a) $R_{rs}(555)$ as a function of $R_{rs}(443)$ for the synthetic dataset (colored data points) and in situ dataset (colored
contours). (b) Same as panel (a) but for the synthetic dataset of Nechad et al. (2015) that was developed for coastal waters. The
color coding refers to the optical water groups as indicated.

For comparing the AOPs from the synthetic database with in situ data, we have chosen two AOPs, the spectral
remote-sensing reflectance, $R_{rs}(\lambda)$, and the spectral diffuse attenuation coefficient of downwelling plane irradiance
averaged within the water column from the sea surface to the first attenuation depth, $<K_d(\lambda)>_1$, and the maximum
band ratio of reflectance, MBR. The scatter plot of our synthetic data of $R_{rs}(555)$ vs. $R_{rs}(443)$ are depicted in Fig.
9a. For comparison, the range of in situ data is illustrated by the dashed contour lines. The maximum values of
$R_{rs}(443)$ reached 0.0165 sr$^{-1}$, which is in good agreement with in situ measurements performed in ultraoligotrophic
waters in the South Pacific gyre during the BIOSOPE cruise (see Fig. 3 in Stramski et al., 2008). These results are
once again illustrated using color coding to represent different optical water types, specifically Groups 1, 2, and 3.
As seen, there is a relatively good agreement between the synthetic data and the range of variability of the in situ
data for Groups 2 and 3 (Fig. 9a). For Group 1 (very turbid waters), however, the synthetic data exhibit a smaller
range of variability compared with in situ data. This result is not unexpected because our primary goal was to
generate the synthetic database that is most representative of open ocean pelagic environments as well as coastal
areas where water turbidity is low to moderate rather than very high. As described in section 2, turbid waters of
Group 1 correspond to Optical Water Classes 1 and 2 as defined in Mélin and Vantrepotte (2015). It is interesting
to note that the synthetic optical database that was developed by Nechad et al. (2015) for coastal waters shows a
relatively good consistency between the synthetic and in situ data for Group 1 (Fig. 8b). However, in contrast to
our synthetic database, the synthetic data of Nechad et al. (2015) exhibit a limited range of variability compared
with in situ data for Groups 2 and 3. Thus, the synthetic data of Nechad et al. (2015) for turbid waters in Group 1
can provide useful complementarity to our synthetic database whose main focus is on water types from Groups 2
and 3.
The scatter plot of the synthetic data of $<K_d(490)>_1$ as a function of blue-to-green band ratio of reflectance,
$R_{rs}(490)/R_{rs}(555)$, is shown in Fig. 10. These synthetic data are again color coded according to optical water classes
defined in terms of Groups, 1, 2, and 3. For comparison, a few empirical relationships between these AOP variables
established in previous analyses of field measurements are also displayed in Fig. 10 (Mueller, 2000; Werdell, 2005;
Werdell, 2009). The relationship of Mueller (2000) was formulated during the early phase of SeaWiFS satellite
mission to serve as an operational global algorithm for estimating $K_d(490)$ from ocean color observations. Werdell
(2005) provided an updated relationship with a primary goal to improve the estimation of $K_d(490)$ at low values
of $K_d(490)$ that correspond to high values of reflectance band ratio. Figure 10 shows that these two relationships
are generally consistent with our synthetic data across the entire range of variability encompassing data from
Groups, 1, 2, and 3. This is reassuring given that the main purpose of our synthetic database and these two empirical
relationships is similar in a sense of targeting the optical variability within the global ocean dominated by open
ocean environments. Figure 10 also includes the relationship of Werdell (2009) that represents the most recent
update of global empirical algorithms for estimating $K_d(490)$ from different ocean color satellite sensors.
Specifically, the relationship of Werdell (2009) presented in Fig. 9 is referred to as KD2S and is based on SeaWiFS
spectral bands. In contrast to relationships of Mueller (2020) and Werdell (2005), the relationship of Werdell
(2009) deviates significantly from our synthetic data within the range of relatively high values of $<K_d(490)>_1$
which correspond to relatively low values of $R_{rs}(490)/R_{rs}(555)$. It is remarkable that this deviation occurs within
the range where our synthetic data are classified as Group 1, so these are the optical water types associated with
high water turbidity. Another remarkable result illustrated in Fig. 10 is that the relationship of Werdell (2009) in
this range is quite consistent with the main trend observed within the synthetic database of Nechad et al. (2015)
that was developed for coastal environments. This result further supports the potential complementarity between
our synthetic database and database of Nechad et al. (2015).

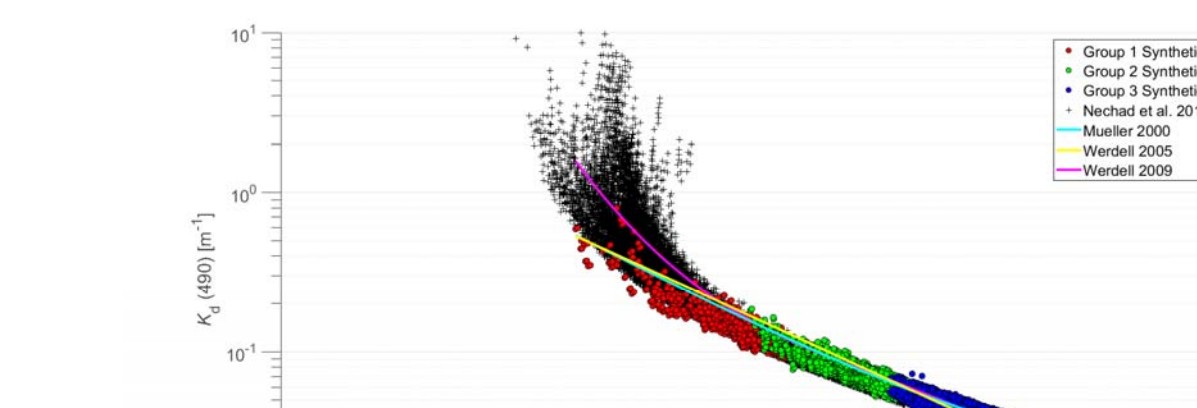


Figure 10. Scatter plot of $K_d(490)$ vs. the blue-to-green reflectance ratio, $R_{rs}(490)/R_{rs}(555)$, for the synthetic database. The red,
green, and blue data points represent the three optical water groups 1, 2, and 3, respectively. The black cross-mark data points
are from the Nechad et al. (2015) synthetic dataset. The curves representing the relationships developed by Mueller (2020),
Werdell (2005), and Werdell (2009) are also displayed. The $K_d(490)$ data points represent $<K_d(490)>_1$ for the present synthetic
database (colored data points), and the near-surface $K_d(490)$ calculated within the top 1 cm layer for the Nechad dataset (black
data points).

The scatter plot of Chla vs. the maximum band ratio of reflectance, MBR, for the synthetic database is shown
in Fig. 11. The monotonically decreasing trend of Chla with increasing MBR is consistent with the SeaWiFS-
specific OC5 algorithm for estimating Chla from MBR (O'Reilly and Werdell, 2019). For this illustration, we
estimated Chla using the relationship between $a_{ph}(660)$ and Chla from Bricaud et al. (1998), which is unavoidably
affected to some extent by natural variability in this relationship.

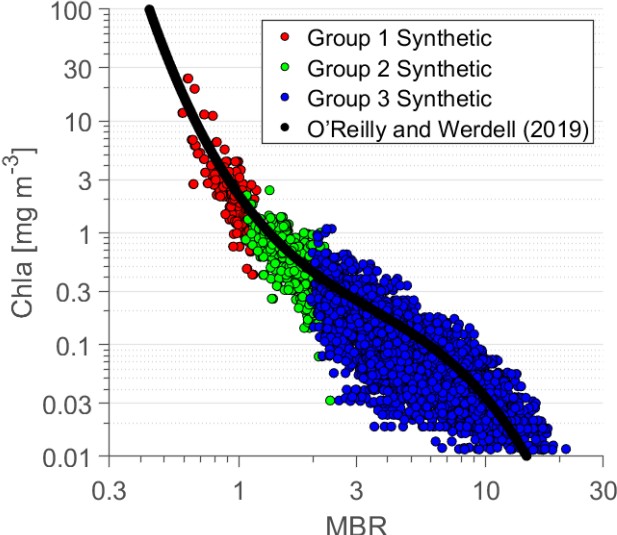


Figure 11. Scatter plot of Chla vs. the blue-to-green maximum band ratio (MBR) of remote-sensing reflectance (i.e,
$R_{rs}(412 > 443 > 490 > 510)/R_{rs}(555)$) for the synthetic database. The red, green, and blue data points represent the three optical
water groups 1, 2, and 3, respectively. The solid black line represents the OC5 algorithm developed by O'Reilly and Werdell
(2019) for SeaWiFS spectral bands. For this illustration, Chla was calculated from $a_{ph}(660)$ using the chlorophyll-specific
phytoplankton absorption at 660 nm from Bricaud et al. (1998).

**5 Summary**
We have generated a new synthetic database that consists of seawater IOPs as well as corresponding
radiometric quantities and AOPs within the ocean surface layer down to a depth of 50 m and at the sea surface.
The radiometric quantities and AOPs were obtained from radiative transfer (RT) simulations performed with
Hydrolight code using the IOPs as input to the calculations. The list of variables included in the database is
provided in Table 2. Because of the use of absorption and scattering properties of pure seawater (assuming the
salinity of 35‰) in the simulations, the present database cannot be used for applications to freshwater
environments and also special caution should be exercised for applications when water salinity is significantly less
than 35‰ because of decrease in pure seawater scattering. This database is organized following an easy to read
netcdf structure and divided into two subsets of data for which the file name identifies the sun zenith angle and the
RT simulation scenario related to the presence or absence of inelastic radiative processes within the water column.
The first subset of data includes the seawater spectral absorption and backscattering coefficients as well as sea-
surface radiometric quantities relevant to ocean color radiometry, $R_{rs}(\lambda)$, $L_w(\lambda)$, $E_d(z=0^+, \lambda)$, and $L_u(z=0^+, \lambda)$ where
$z=0^+$ is just above the surface. The surface and depth-profile values of several spectral radiometric quantities and
AOPs, as well as *PAR* are included in the second subset of data. The spectra of $z_{eu}$ and $z_1$ are also provided in the
second file. More details on the organization and content of the database are included in readme file that is also
provided in the database.
In closing, we present an example illustration of one of the radiometric variables included in the output data
files generated by RT simulations. We recall that the primary result of HydroLight simulations is the spectral
radiance that provides a comprehensive information about the angular distribution of light field, from which
different irradiances and AOPs are calculated. However, it is the spectral downwelling plane irradiance, $E_d(z, \lambda)$,
that has been the most commonly measured radiometric quantity in ocean optics, so in Fig. 12 we have chosen to
illustrate the HydroLight-simulated $E_d(z, \lambda)$ within the ocean surface layer down to a depth of 50 m. These results
are presented for three different scenarios of IOPs which are representative of three different optical water types
defined in terms of Group 1, Group 2, and Group 3 (see section 2). These RT simulations were performed for the
sun zenith angle of 30° in the presence of Raman scattering by water molecules and chlorophyll-a fluorescence in
the water column. In addition to significant differences in the variation of the spectral $E_d(\lambda)$ as a function of depth
$z$ between the Groups 1, 2, and 3, Fig. 12 also illustrates distinct differences in the magnitude and spectral behavior
of the first optical attenuation depth, $z_1$. This quantity is equivalent to the inverse of the diffuse attenuation
coefficient, $\langle K_d(\lambda) \rangle_1$. As expected, the first attenuation depth $z_1$ is located much closer to the ocean surface for
data from Group 1 (Fig. 12a) compared with Group 2 (Fig. 12b) and Group 3 (Fig. 12c), especially across the blue-
green region of the spectrum. In the red part of the spectrum where pure water absorption dominates the attenuation
of $E_d(\lambda)$, the differences between the three groups are small. It is also notable that the spectral behavior of $z_1$ for
Group 3 (Fig. 12c) that represents relatively clear ocean waters is remarkably similar to the spectral shape of pure
water absorption coefficient.

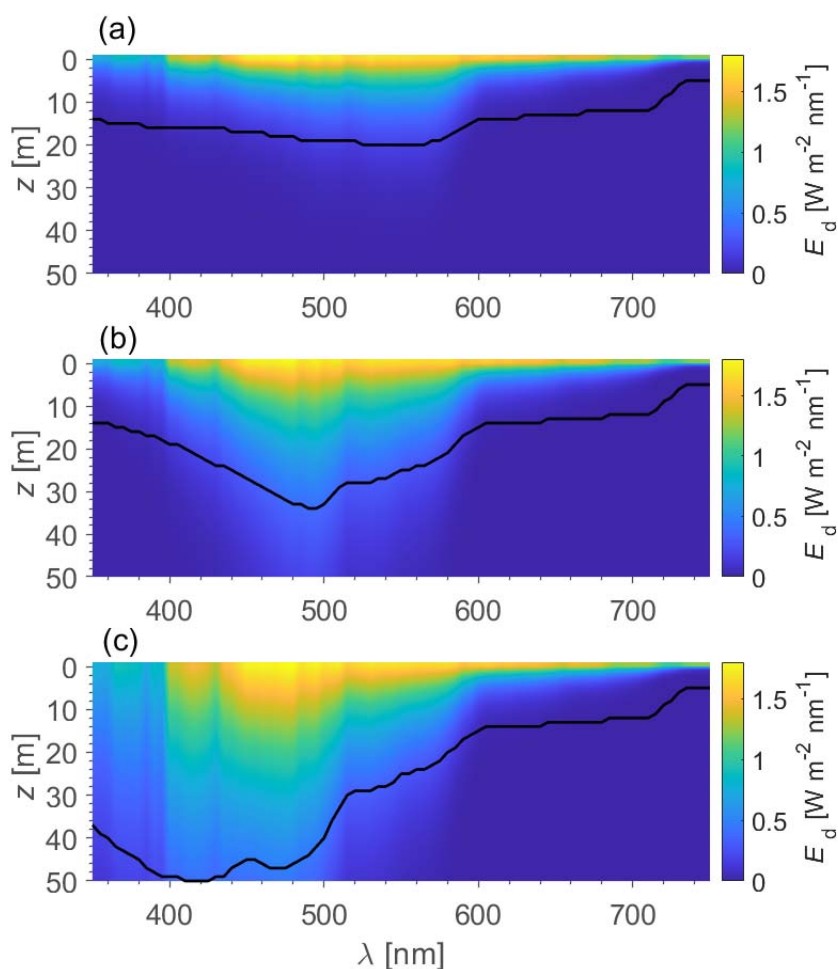

Figure 12. Examples of depth profiles of $E_d(z, \lambda)$ for a given IOP scenario from (a) the optical water group (OWG) 1, (b)
OWG 2, and (c) OWG 3. Radiative transfer simulations were performed for a sun zenith angle of 30° and included Raman
scattering by water molecules and chlorophyll-a fluorescence.

**Table 2:** Symbols, variables, and units for the various quantities included in the final synthetic optical database.

| Symbol | Variable* | Units |
|---|---|---|
| $z$ | Depth in water | m |
| $\lambda$ | Light wavelength in vacuum | nm |
| $a$, $b$, $b_b$ | Total absorption, scattering, and backscattering coefficients of seawater | $m^{-1}$ |
| $a_{nw}$ | Absorption coefficient of all non-water constituents | $m^{-1}$ |
| $a_{ph}$, $a_d$, $a_g$ | Absorption coefficients of phytoplankton, non-algal particles, and CDOM | $m^{-1}$ |
| $b_{nw}$ | Backscattering coefficient of all non-water constituents | $m^{-1}$ |
| $b_{b\text{-}ph}$, $b_{b\text{-}d}$ | Backscattering coefficients of phytoplankton and non-algal particles | $m^{-1}$ |
| $b_{nw}$ | Scattering coefficient of all non-water constituents | $m^{-1}$ |
| $b_{ph}$, $b_d$ | Scattering coefficients of phytoplankton and non-algal particles | $m^{-1}$ |
| $E_o$, $E_{od}$, $E_{ou}$ | Total, downwelling, and upwelling scalar irradiances | $W\ m^{-2}\ nm^{-1}$ |
| $E_d$, $E_u$ | Downwelling and upwelling plane irradiances | $W\ m^{-2}\ nm^{-1}$ |
| $L_w$, $L_u$ | Water-leaving and upwelling radiances | $W\ m^{-2}\ sr^{-1}\ nm^{-1}$ |
| $PAR$ | Photosynthetically Available Radiation defined as the total quantum scalar irradiance within the spectral range 400-700 nm | µmol photons $s^{-1}\ m^{-2}$ |
| $R_{rs}$ | Remote-sensing reflectance | $sr^{-1}$ |
| $K_x$ | Diffuse attenuation coefficients for upwelling and downwelling plane irradiances or upwelling radiance (the radiometric quantity is indicated by subscript x) | $m^{-1}$ |
| $\mu_d$, $\mu_u$ | Average cosines of downwelling and upwelling light fields | dimensionless |
| $z_{eu}$ | Euphotic depth at which $PAR$ is reduced to 1% of its surface value | m |
| $z_1$ | First optical attenuation depth at which spectral $E_d$ or $PAR$ is reduced to 36.8% of its surface value | m |

*All optical variables in the database are spectral and provided at different light wavelengths between 350 and
750 nm at 5 nm intervals and different depths within the water column between the sea surface and the 50 m
depth, except for $R_{rs}$, and $L_w$ which are defined at the sea surface.

**Author contributions.** The concept of this study originated from the authors' discussions about the need for a
new synthetic optical database in support of ocean color science and applications, especially the global ocean
applications, including support of upcoming NASA's PACE hyperspectral ocean color satellite mission. All co-
authors contributed to curation of in situ data. HL and DSFJ led the generation of the synthetic IOP dataset and
created the satellite IOP dataset. DSFJ ran the RT simulations. HL and DS wrote the manuscript. All co-authors
contributed to discussion, review, and editing of the manuscript.
**Competing interests.** The authors declare that they have no conflict of interest.
**Disclaimer.** Mention of trade names or commercial products does not constitute endorsement or recommendation
for use. The views expressed in this article are those of the authors.
**Acknowledgements.** We gratefully acknowledge all scientists and supporting personnel involved in collection,
processing, and dissemination of in situ and satellite data used in this study as well as all agencies that provided
support for these activities. We thank Jérôme Vialard for the generation of global SST data. We also thank Jaime
Pitarch and two anonymous reviewers for comments on the manuscript.
**Data Availability**: The DOI (doi:10.6076/D1630T) is not yet active but reserved at the Dryad data repository. The
database is available at Dryad open-access repository of research data (Loisel et al., 2023). Following completion
of the review process,
the synthetic optical database described in this study will be
publicly available at the Dryad open-access repository of
research data (Loisel et al., 2023; https://doi.org/10.6076/D1630T).
**Financial support.** This study was supported by the ANR CO2COAST project (ANR-20-CE01-0021 awarded to
Hubert Loisel) and the National Aeronautics and Space Administration in USA through the PACE project (NASA
Grant 80NSSC20M0252 awarded to Dariusz Stramski and Rick. A. Reynolds).
**Review statement.**

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
