# Peer review of "A synthetic database generated by radiative transfer simulations in support of studies in ocean optics and optical remote sensing of the global ocean"

_Earth System Science Data, 2023_

## Referee Comment (RC1)

**Review of the manuscript "A synthetic database generated by radiative transfer simulations in support of studies in ocean optics and optical remote sensing of the global ocean", by Loisel et al.**

**General comments**

This manuscript describes the construction of a synthetic dataset for optical studies in the ocean, using Hydrolight. This topic is very familiar to me right now as I am pursuing a similar goal, so it was an easy read. Authors must make sure that it is accessible to a broader audience though.

It proceeds the usual way, as in the old IOCCG dataset from Lee in 2003: first it assembles a set of phytoplankton absorption spectra, then the rest of IOPs are built with relationships that include some randomness. Finally, a single wind speed (5 m/s) and three sun angles (0°, 30° and 60°) are set, as well as various combinations of inelastic scattering on and off. I downloaded and saw the dataset as part of the review.

Things I liked:

- The randomness in the bio-optical relationships, that will reproduce the spread in the relationships that is observed in nature.
- The Petzold phase function is abandoned and the much more realistic Fournier-Forand is considered for non-algal particles. Maybe a remark by the authors would be better.
- The 50 nm gap left for Raman scattering. In fact, I checked with my own simulations that the spectral memory of Raman scattering is about 50 nm, so it makes sense. A comment by the authors would be appreciated.
- The organization in netcdf files is quite handy compared to the Hydrolight text files.

Now I have a list of things I liked less:

I have made a ternary plot of the absorption budget and I have compared it with the IOCCG (Lee) and the Coastcolour (Nechad) datasets. What I see here is a disproportionately low amount of non-algal particles, even compared to the IOCCG dataset, which was developed for ocean applications. I am not saying that IOCCG is right and this one is wrong, but authors should verify that such absorption budget is what it is actually found in the global oceans. Compared to other datasets, $b_b$ appears lower too.

[Figure]

I have also plotted the remote-sensing reflectances (no inelastic scattering, sun at 30°):

[Figure]

Some $R_{rs}$ look crazy for me. I have never seen anything that high in the blue, even for the most oligotrophic waters. To verify, I have calculated the maximum band ratio (MBR) and I have applied the OC4 to it, according to O'Reilly and Werdell (2019). I have also calculated the chlorophyll index (CI), by Hu et al. (2012), for the most oligotrophic waters and I have applied his algorithm too. I get two chlorophyll histograms for the whole dataset:

[Figure]

[Figure]

Considering that the lowest CHL measured in Valente et al. (2019), cited in O'Reilly and Werdell (2019), was 0.012 mg m$^{-3}$, that leaves us a very high amount of simulations whose CHL is unlikely low, whether we use OC4 or CI (Hu) to compare with. I also checked with Morel "clearest" waters and these values are definitely off. I therefore encourage redefinition of the dataset. I do not have an explanation for this artifact considering that the authors have reproduced the histograms seen by satellite data. I can hypothetise (1) the retrievals were biased the $a_{ph}(440)$ is actually higher or (2) the bio-optical relationships affect the CHL algorithm and need redefinition.

Related to this, there are datasets that may help in getting bio-optical relationships that are realistic. For example, I compared some absorption ratios to NOMAD:

[Figure]

I think I see that for the same $a_{ph}(440)$, there is a general lower value for $a_g(440)$ compared to NOMAD. Regarding $a_d(440)$, I see a lack of spread.

This is not the only example of what the authors can do. For example, I have plotted the CDOM slope $S_g$ as a function of $a_g(440)$ for the NOMAD and Biosope datasets, as well as for three cruises in very clears waters of our group:

[Figure]

One can see some tendency to spread, especially to high $S_g$, when $a_g(440)$ is small, and a tendency to concentrate around Sg≈0.016 $nm^{-1}$ for high $a_g(440)$. But the authors use a uniform distribution between 0.01 $nm^{-1}$ and 0.02 $nm^{-1}$. This could therefore be improved.

I could revise the rest of IOPs and bio-optical relationships but I believe that at this point the authors got my point.

**Specific comments**

Abstract: it lacks a motivation on why another dataset is needed

Lines 51-52: "Recent technological developments and broader accessibility of optical in situ instrumentation" I believe this is unfortunately not the case. Seabird (old Wetlabs and Satlantic) has discontinued many in situ optical instrument, HobiLabs has closed and is not selling instruments anymore. All we have is Sequoia and Seabird in a situation of monopoly with little or no incentive to innovate and imposing high prices in already old design instruments, with a general lack of market competition.

Lines 60-63: the most important motivation for a synthetic dataset is that we will never have complete optical datasets across the widest dynamic range, and with declared and low uncertainties.

Lines 118:120: this is unclear to me.

Line 145: I would avoid the word "specific" as it is usually referred to the absorption divided by the concentration.

Lines 152-153: I think all IOPs matter equally, not only $a_{ph}$.

Line 160: "the measured values of $a_{ph}(\lambda)$ were used in the calculations of these IOPs". Alright, but Lee did the same 20 years ago, so it is not a big novelty. I would not emphasize.

Lines 238-241: this comment is totally right. In fact, it is a pity that in 2023 there are still new datasets that are degrading spectral resolution to only few bands. Not to mention the aggregation of $a_g$ and $a_d$ in Valente, which makes us still rely on NOMAD when we want them separately.

On the reconstruction of hyperspectral $a_{ph}$ from multispectral: I believe that a decently sized of hyperspectral $a_{ph}$ can be compiled without the need to worry about this.

Line 276: When extrapolating $a_{ph}$ to the UV, how is exactly the UV part "glued" to the rest?

Line 311: probably instead of "shifted", I would say "biased".

Lines 345-346: I think it is stated that the Mediterranean Sea is ultraoligotrophic, when it is actually not, not even the eastern basin (maybe this place in Summer, yes).

Line 460: "$m^2$/(mg Chla)". Mass is mass, so please delete the "Chla". Yes, it is common to write it like that among some biologists, but it does not make sense metrologically.

Lines 460-461: it is much more accurate to use a red wavelength of $a_{ph}$ rather than a blue one to estimate CHL.

Lines 536-540: I wonder what are the reason to not consider the pure water measurements by Mason and Fry in 2016.

Lines 551-553: I wonder whether saving the whole profile is very useful, considering that Hydrolight already calculates for you the "K's", "z's" and these depth-related quantities.

Figure 7 is not an efficient way to show the differences. Of course, everything increases with $a_{ph}(440)$ to a first order, but we want to know the differences among datasets. I prefer if the ratios are represented e.g., $a_g(440)/a_{ph}(440)$ as a function of aph(443), etc.

Line 602: "The scatter plots show a significant degree of overlap" Very roughly, but see my comment above.

Lines 688-689: there is no complementarity of this dataset and Nechad's as both have different assumptions regarding the bio-optical modelling, so they are not consistent with each other.

The plots in Fig. 10 are not telling anything new as we know what happens with $E_d$ profiles for different water types.

---

## Author Comment (AC1)

**Responses to Reviewer #1**

"A synthetic optical database generated by radiative transfer simulations in support of studies in ocean optics and optical remote sensing of the global ocean"

Hubert Loisel, Daniel Schaffer Ferreira Jorge, Rick A. Reynolds, and Dariusz Stramski

We appreciate the constructive comments by Dr. Pitarch. Here we provide our detailed point-by-point responses and a description of any actions taken in regard to these comments. The Reviewer's comments are shown in italicized font; our responses follow each comment in normal font. Line numbers and figures indicated in our responses refer to the revised manuscript unless otherwise noted.

*General comments. This manuscript describes the construction of a synthetic dataset for optical studies in the ocean, using Hydrolight. This topic is very familiar to me right now as I am pursuing a similar goal, so it was an easy read. Authors must make sure that it is accessible to a broader audience though.*
*It proceeds the usual way, as in the old IOCCG dataset from Lee in 2003: first it assembles a set of phytoplankton absorption spectra, then the rest of IOPs are built with relationships that include some randomness. Finally, a single wind speed (5 m/s) and three sun angles (0°, 30° and 60°) are set, as well as various combinations of inelastic scattering on and off. I downloaded and saw the dataset as part of the review.*
*Things I liked:*
- *The randomness in the bio-optical relationships, that will reproduce the spread in the relationships that is observed in nature.*
- *The Petzold phase function is abandoned and the much more realistic Fournier-Forand is considered for non-algal particles. Maybe a remark by the authors would be better.*
- *The 50 nm gap left for Raman scattering. In fact, I checked with my own simulations that the spectral memory of Raman scattering is about 50 nm, so it makes sense. A comment by the authors would be appreciated.*
- *The organization in netcdf files is quite handy compared to the Hydrolight text files.*

**Response:** We thank the Reviewer for positive comments on our manuscript. Regarding Raman scattering the center of the emission band $\lambda_{em}$ is related to the center of the excitation wavelength $\lambda_{ex}$ through the following equation: $\lambda_{em} = 10^7/(10^7/\lambda_{ex} - 3400)$. The wavelength shift is around 50 nm for excitation in the UV-blue but increases to >100 nm for excitation in the red. This is described in Mobley (2012) which is cited in the manuscript where the wavelength distribution function is specified. We think there is no need for more detailed description of Raman scattering in our manuscript.

*Now I have a list of things I liked less. I have made a ternary plot of the absorption budget and I have compared it with the IOCCG (Lee) and the Coastcolour (Nechad) datasets. What I see here is a disproportionately low amount of non-algal particles, even compared to the IOCCG dataset, which was developed for ocean applications. I am not saying that IOCCG is right and this one is wrong, but authors should verify that such absorption budget is what it is actually found in the global oceans. Compared to other datasets, $b_b$ appears lower too.*

**Response:** We agree with the Reviewer's comment that the absorption dataset in our orginal manuscript is generally characterized by a lower contribution of non-algal particles in the absorption budget compared to the IOCCG and CoastColour datasets. We note that comparison with the CoastColour dataset, however, is not fully pertinent as that dataset was developed specifically for coastal waters with relatively high contribution of non-algal particles. Taking the Reviewer's comment into account, however, we compared our original dataset with in situ data from open ocean waters (BIOSOPE cruise in the subtropical Pacific Ocean and the Atlantic data points from the CoastlOOC project). We concluded that it indeed made sense to regenerate our absorption dataset de novo allowing for an increased proportion of non-algal particles. To

accomplish this, the parameter $P_2$ in Eq. 3 was changed (see new equation in Table 1 of revised manuscript) to allow the contribution of $a_d$ to vary between 10 and 90% of $a_{ph}$. The main implication of creating this new absorption dataset for the revised manuscript is that we regenerated the entire synthetic optical database presented in our study by rerunning all Hydrolight radiative transfer simulations for the same simulation scenarios as presented in the original manuscript. Accordingly, multiple figures were replaced with revised figures representing the new database (i.e., Figs. 5, 7–11). In general, however, the recalculation of the database did not require significant revisions of the text although some revisions and edits, where appropriate, were made. This new database of simulations will also replace the original database on the publicly-available Dryad repository.

Figure R1-1 (below) presents a comparison of the original and revised absorption coefficients in ternary plots, as proposed by the Reviewer. This comparison shows that the new absorption dataset (right panel) yields an absorption budget that includes higher contributions of $a_d$ and is more consistent with in situ measurements collected in open ocean waters.

[Figure]

Fig. R1-1 (not in the manuscript). Ternary plots of the absorption dataset in the original manuscript (a) and the new absorption dataset used in the revised manuscript (b). Black data points represent the synthetic dataset and red data points are in situ measurements from open ocean waters in the Pacific (BIOSOPE) and Atlantic (CoastlOOC).

With regards to the comment that "*Compared to other datasets, $b_b$ appears lower too*", this is generally true. Similar to Fig. 7c in our manuscript, Fig. R1-2 (below) shows the $b_{bp}$ vs. $a_{ph}$ relationships of Huot et al. (2008) and Antoine et al. (2011) established from in situ measurements in open ocean waters and compares them with the synthetic datasets from IOCCG, Craig et al. (2020), and our present work. Figure R1-2 clearly indicates that, for a given $a_{ph}$ (or Chla), $b_{bp}$ is overestimated in both the IOCCG and Craig et al. datasets, especially in oligotrophic waters, which is mainly due to the high contribution of non-algal particles in these datasets. In contrast, the distribution of data points in our dataset (left panel) is consistent with the main trend lines from Huot et al. (2008) and Antoine et al. (2011).

We added new text describing these results in the revised manuscript (lines 593-600). We did not add any new figure on this specific aspect except for a new panel (c) in Fig. 7 showing $a_d$(443) vs. $a_{ph}$(443) for our synthetic and in situ datasets. Discussion about the ratio of different IOPs to $a_{ph}$(443) vs. $a_{ph}$(443) has been also refined (lines 593-600).

[Figure]

Fig. R1-2 (not in the manuscript). Comparison of $b_{bp}(550)$ as a function of $a_{ph}(443)$ for different synthetic datasets. Empirical relationships describing $b_{bp}(550)$ vs. Chla reported from field measurements are also displayed for comparison. For these latter relationships, $a_{ph}(443)$ has been estimated from Chla as described in the manuscript.

*I have also plotted the remote-sensing reflectances (no inelastic scattering, sun at 30°). Some Rrs look crazy for me. I have never seen anything that high in the blue, even for the most oligotrophic waters. To verify, I have calculated the maximum band ratio (MBR) and I have applied the OC4 to it, according to O'Reilly and Werdell (2019). I have also calculated the chlorophyll index (CI), by Hu et al. (2012), for the most oligotrophic waters and I have applied his algorithm too. I get two chlorophyll histograms for the whole dataset: Considering that the lowest CHL measured in Valente et al. (2019), cited in O'Reilly and Werdell (2019), was 0.012 mg m-3, that leaves us a very high amount of simulations whose CHL is unlikely low, whether we use OC4 or CI (Hu) to compare with. I also checked with Morel "clearest" waters and these values are definitely off. I therefore encourage redefinition of the dataset. I do not have an explanation for this artifact considering that the authors have reproduced the histograms seen by satellite data. I can hypothetise (1) the retrievals were biased the aph(440) is actually higher or (2) the bio-optical relationships affect the CHL algorithm and need redefinition.*

**Response:** The maximum values of $R_{rs}$ spectrum which have been measured in open ocean waters can reach about 0.026 sr$^{-1}$ in the violet-blue part of the spectrum (see, for example, BIOSOPE measurements in Fig. 3 of Stramski et al., 2008). In our original manuscript, only 0.0636% of $R_{rs}$ spectra had values higher than 0.026 sr$^{-1}$. The Chla histograms provided by the Reviewer show also only a very small fraction of Chla data lower than 0.012 mg m$^{-3}$ (note that for such very clear waters, only the Hu et al. algorithm should be considered). We verified that in our newly simulated database presented in the revised manuscript, the $R_{rs}$ values higher than 0.026 sr$^{-1}$ do not exist in the configuration mentioned in the Reviewer's comment (i.e., no inelastic scattering and sun at 30°). When all our new simulations are considered, the maximum value of $R_{rs}$ reaches 0.0029 sr$^{-1}$, and only 7 out of 29880 spectra (0.023%) in our new synthetic database have values higher than 0.026 sr$^{-1}$ (see Fig. R1-3 below). We find these results completely reasonable and adequate. The following addition has been made in the revised manuscript (lines 644-647) where the $R_{rs}(555)$ vs. $R_{rs}(443)$ pattern is described (Fig. 9): "The maximum values of $R_{rs}(443)$ reached 0.0165 sr$^{-1}$, which is in good agreement with in situ measurements performed in ultraoligotrophic waters in the South Pacific gyre during the BIOSOPE cruise (see Fig. 3 in Stramski et al., 2008)."

[Figure]

Fig. R1-3 (not in the manuscript): The new synthetic dataset of $R_{rs}$ spectra presented and used in the revised manuscript.

*Related to this, there are datasets that may help in getting bio-optical relationships that are realistic. For example, I compared some absorption ratios to NOMAD: I think I see that for the same $a_{ph}(440)$, there is a general lower value for $a_g(440)$ compared to NOMAD. Regarding $a_d(440)$, I see a lack of spread.*

**Response:** It is important to realize that the NOMAD dataset is dominated by coastal waters which explains why for a given $a_{ph}$, $a_d$ and $a_g$ values often tend to be higher in NOMAD compared to typical range of scenarios expected for open ocean waters (the main focus of our study). The lack of spread observed for $a_d$ has now been corrected in the revised manuscript as described in our previous responses. By plotting the $a_g/a_{ph}$ as a function of $a_{ph}$ (Fig. R1-4 below) one may note a relatively good overlap between in situ and synthetic data (although admittedly in situ data of $a_g$ are quite scarce in very clear waters).

[Figure]

Fig. R1-4 (not in the manuscript). The ratio $a_g/a_{ph}$ as a function of $a_{ph}$ for the synthetic (colored points) and in situ (black points) datasets.

*This is not the only example of what the authors can do. For example, I have plotted the CDOM slope Sg as a function of ag(440) for the NOMAD and Biosope datasets, as well as for three cruises in very clears waters of our group. One can see some tendency to spread, especially to high Sg, when ag(440) is small, and a tendency to concentrate around Sg≈0.016 nm-1 for high ag(440). But the authors use a uniform*

*distribution between 0.01 nm-1 and 0.02 nm-1. This could therefore be improved. I could revise the rest of IOPs and bio-optical relationships but I believe that at this point the authors got my point.*

**Response:** We prefer to keep a uniform distribution of $S_g$ for two reasons. First, a direct link between $S_g$ and $a_g$ is not well established and it is difficult to provide an average trend between these two parameters. Second, the whole spectra of IOPs are provided in our database, so the $S_g$ slope can be computed for every $a_g$ spectrum which provides users with the freedom to select cases of interest, if deemed appropriate for specific studies.

*Specific comments*
*Abstract: it lacks a motivation on why another dataset is needed*

**Response:** The abstract already states that "Compared to similar developments of optical databases in the past, the present dataset of IOPs is characterized by probability distributions of IOPs that are consistent with global distributions representative of vast areas of open ocean pelagic environments and coastal regions covering a broad range of optical water types". We also mention that the new optical database obtained from radiative transfer simulations accounts for inelastic scattering, which is not the case in previous databases. We made minor edits to further clarify this point in the abstract (lines 28-29): "These input IOPs were used in three simulation scenarios associated with assumptions about inelastic radiative processes (not considered in previous synthetically-generated optical databases) in the water column…."

*Lines 51-52: "Recent technological developments and broader accessibility of optical in situ instrumentation" I believe this is unfortunately not the case. Seabird (old Wetlabs and Satlantic) has discontinued many in situ optical instrument, HobiLabs has closed and is not selling instruments anymore. All we have is Sequoia and Seabird in a situation of monopoly with little or no incentive to innovate and imposing high prices in already old design instruments, with a general lack of market competition.*

**Response:** We do not wish to enter this discussion in any greater detail and it is beyond the main thrust of our study. While it is true that some commercial products have been discontinued or may be discontinued soon, it is also true that technological advancements are underway (e.g., under current SBIR programs in the US) both in terms of radiometric and IOP instrumentation and some new instruments have already reached either the state of commercialization or will likely get to this point in near future (e.g., Sunstone Scientific). Also, there are still some relevant commercial products available from a few other companies such as Biospherical, TriOS, CIMEL, or RBR.

*Lines 60-63: the most important motivation for a synthetic dataset is that we will never have complete optical datasets across the widest dynamic range, and with declared and low uncertainties.*

**Response:** We agree that field datasets are unlikely to fulfill this desire for complete optical datasets spanning the entire dynamic range of the ocean. Similarly, synthetic optical databases are unlikely to be ever developed to the point to cover all possible natural conditions in the ocean, primarily because such databases depend on the use of simplified input parameters characterizing the complex environmental variability. For example, presently the variability in particle phase function can be viewed as imposing some limitations. One important point about the synthetic databases is that the data are free of measurement errors, so we made minor edits to better clarify this point (line 62): "In this context, radiative transfer (RT) simulations, which are free of measurement errors, provide a useful tool to generate comprehensive synthetic databases and complement the existing datasets of field measurements in support of studies in ocean optics and optical remote sensing."

*Lines 118:120: this is unclear to me.*

**Response:** Part of the original sentence, which is not the most important point, has been removed. The sentence now reads (line 117): "Third, the probability distributions of different IOPs that were used as input to previous RT simulations do not appear to match well with the IOP distributions observed in extensive field datasets or satellite-derived datasets representing the global ocean."

*Line 145: I would avoid the word "specific" as it is usually referred to the absorption divided by the concentration.*

**Response:** This has been changed and reformulated as follows: "Specifically, the absorption coefficients of the different constituents are the spectral absorption coefficients of phytoplankton, ......". The same modification has been made in other places where relevant.

*Lines 152-153: I think all IOPs matter equally, not only $a_{ph}$.*

**Response:** We agree that all IOPs matter equally, but here for the creation of the synthetic dataset $a_{ph}$ is used as a main "driver" to define and constrain the variability of other IOPs that are expected to occur in open ocean waters. In the manuscript we have stated (lines 155-156): "Among these different constituent IOPs, the phytoplankton absorption coefficient, $a_{ph}(\lambda)$, plays the most fundamental role in the creation of the synthetic dataset of IOPs in this study."

*Line 160: "the measured values of $a_{ph}(\lambda)$ were used in the calculations of these IOPs". Alright, but Lee did the same 20 years ago, so it is not a big novelty. I would not emphasize.*

**Response:** We believe it is important to keep this sentence because it indicates that our IOP dataset has been generated based on in situ measurements of $a_{ph}$, which was not the case for the IOCCG dataset. In this context, our main purpose is to emphasize this point rather than address any particular novel aspects compared to previous synthetic datasets.

*Lines 238-241: this comment is totally right. In fact, it is a pity that in 2023 there are still new datasets that are degrading spectral resolution to only few bands. Not to mention the aggregation of $a_g$ and $a_d$ in Valente, which makes us still rely on NOMAD when we want them separately. On the reconstruction of hyperspectral $a_{ph}$ from multispectral: I believe that a decently sized of hyperspectral $a_{ph}$ can be compiled without the need to worry about this.*

**Response:** We thank the Reviewer for this comment.

*Line 276: When extrapolating $a_{ph}$ to the UV, how is exactly the UV part "glued" to the rest?*

**Response:** Once the reference spectrum that exhibits the best correlation with the investigated spectrum in the visible has been identified, the UV portion of the reference spectrum is normalized to its value at 400 nm. This normalized UV spectrum is then multiplied by the reference spectrum value at 400 nm and used to extend the investigated spectrum into the UV.

*Line 311: probably instead of "shifted", I would say "biased".*

**Response:** We think "shifted" is more appropriate than "biased" in this context. The datasets that contain significant fraction of coastal measurements are not necessarily biased but their statistical measures of central tendency are shifted to larger values compared to predominantly open ocean data.

*Lines 345-346: I think it is stated that the Mediterranean Sea is ultraoligotrophic, when it is actually not, not even the eastern basin (maybe this place in Summer, yes).*

**Response:** We agree that the Mediterranean Sea is not ultraoligotrophic as a whole. The Loisel et al. (2011) paper refers to some ultraoligotrophic eddies within the Mediterranean Sea observed during summer. This point has been clarified (lines 357-361): "While the original classification of Mélin and Vantrepotte (2015) includes 16 optical water classes (OWC), the derivation of $a_{ph}(\lambda)$ and $a_{dg}(\lambda)$ from the 3SAA additionally included a $17^{th}$ OWC to improve the representation of ultraoligotrophic waters such as those found in the South Pacific Gyre (Morel et al., 2007; Claustre et al., 2008; Stramski et al., 2008) and in some areas of the Mediterranean Sea in summer (Loisel et al., 2011). This $17^{th}$ OWC is described in Jorge et al. (2021)."

*Line 460: "m2/(mg Chla)". Mass is mass, so please delete the "Chla". Yes, it is common to write it like that among some biologists, but it does not make sense metrologically.*

**Response:** "Chla" has been removed.

*Lines 460-461: it is much more accurate to use a red wavelength of $a_{ph}$ rather than a blue one to estimate CHL.*

**Response:** We agree that $a_{ph}$ in the red is a better proxy of Chla because this band is less affected by various accessory pigments and package effect than the blue absorption band. However, in these specific calculations this aspect is not critically important because the purpose is to generate a relatively large range of variability in one of the IOP coefficients, which is accomplished through the use of random factor, and not to predict Chla.

*Lines 536-540: I wonder what are the reason to not consider the pure water measurements by Mason and Fry in 2016.*

**Response:** We use the spectral values of $a_w(\lambda)$ following the current recommendation of the IOCCG (2018) protocols devoted to the absorption coefficient (Table 1.1 in Chapter 1). The Mason and Fry (2016) measurements were not included in these recommendations. The Mason and Fry values are significantly lower in the short-wavelength portion of the spectrum than other literature values which are supported by relevant discussion in the IOCCG protocols. The significantly different values of Mason and Fry have not yet been vetted by the community and would require support from additional studies and validation to reach adequate level of confidence for potential use as "standard" recommended values.

*Lines 551-553: I wonder whether saving the whole profile is very useful, considering that Hydrolight already calculates for you the "K's", "z's" and these depth-related quantities.*

**Response:** Data of whole profile are useful; for example, to calculate an averaged $K_d$ value over a specific water layer which can be of interest to users and some applications.

*Figure 7 is not an efficient way to show the differences. Of course, everything increases with $a_{ph}(440)$ to a first order, but we want to know the differences among datasets. I prefer if the ratios are represented e.g., $a_g(440)/a_{ph}(440)$ as a function of aph(443), etc.*

**Response:** The objective of this figure is to show how the present synthetic dataset compares with in situ data. Similar patterns are observed when IOPs are divided by $a_{ph}$ (see Fig. R1-5 below), which is now mentioned in the text of revised manuscript text (lines 597-599) but without adding a new figure such as Fig. R1-5 in the revised manuscript (which already has 12 figures). In addition, as discussed in Berges (Limnol. Oceanogr. 42, 1006-1007, 1997), plots of Y/X vs. X should be interpreted with special caution.

We also note that in Fig. 7 of the revised manuscript we added a new panel with $a_d(443)$ vs. $a_{ph}(443)$ and polygon lines to improve the illustration of the range of in situ data.

[Figure]

Fig. R1-5 (not in the manuscript): The ratio of constituent absorption coefficients to $a_{ph}$ vs. $a_{ph}$ at 443 nm for the synthetic (colored data points) and in situ (black data points) datasets.

*Line 602: "The scatter plots show a significant degree of overlap" Very roughly, but see my comment above.*

**Response:** We think our response above addresses this point.

*Lines 688-689: there is no complementarity of this dataset and Nechad's as both have different assumptions regarding the bio-optical modelling, so they are not consistent with each other.*

**Response:** These two synthetic datasets have been generated with different assumptions regarding bio-optical modeling as their primary focus is on different bio-optical environments (open vs. coastal waters). So, in that sense we think these datasets are complementary as their combination covers a larger and more diverse range of optical environments.

*The plots in Fig. 10 are not telling anything new as we know what happens with $E_d$ profiles for different water types.*

**Response:** The purpose of this figure is not to provide scientifically novel information but rather to illustrate the spectral and vertically-resolved (along the water column) optical information included in this new synthetic database. This can be useful to readers interested in this kind of optical data, especially that other commonly known synthetic optical databases (IOCCG, 2006; Craig et al., 2020) do not include data as a function of depth within the water column.

---

## Author Comment (AC2)

**Responses to Reviewer #2**

"A synthetic optical database generated by radiative transfer simulations in support of studies in ocean optics and optical remote sensing of the global ocean"

Hubert Loisel, Daniel Schaffer Ferreira Jorge, Rick A. Reynolds, and Dariusz Stramski

We appreciate the constructive comments by the Reviewer. Here we provide our detailed point-by-point responses and a description of any actions taken in regard to these comments. The Reviewer's comments are shown in italicized font; our responses follow each comment in normal font. Line numbers and figures indicated in our responses refer to the revised manuscript unless otherwise noted.

*The manuscript by Loisel and co-workers presents a synthetic data set of so-called apparent optical properties (AOPs) generated through simulation of the radiative transfer (RT) in oceanic waters. The goal is to help the development and test of inversion algorithms that use these AOPs as input (generally the remote-sensing reflectance, Rrs) and attempt to derive quantities such as the inherent optical properties (IOPs, here the absorption and backscattering coefficients) or biogeochemical / ecological properties such as the phytoplankton chlorophyll concentration (Chl). The Rrs are generally derived from satellite ocean colour observations, although the inversion can also be applied to Rrs derived from in-situ radiometry measurements.*

*Numerous such data sets exist and the justification for proposing this new one is that the IOPs used as inputs to the RT simulations are made more representative of the real world by using probability distribution functions that are consistent with what global in-situ or satellite data sets reveal.*

*I guess this is an important feature if the data set is then used in its entirety to, e.g., train ML-based algorithms such as neural networks. In such a case, it is indeed critical that the training data set is as realistic as possible. If the data set is used by expert users who can appropriately select what they need in this simulated data (e.g., they can pick up whatever range of the various input parameters that they think is relevant to their development), then the justification for this new data set is a bit weaker.*

*In any case, I think it is a very useful data set. I appreciate that the authors have included not only the Rrs spectra but also most apparent optical properties as well as the full radiance distributions. This will undoubtedly make this dataset of a broader interest.*

*The effort to make the synthetic data set better representative of the real global ocean, which is largely made of clear and moderately clear waters, is also a very good one. As clearly shown by Fig. 6, previous synthetic data sets did not match at all what the global ocean is. They were largely skewed towards high values of most IOPs, which is rather typical of coastal waters. Still, they were often used to develop and test algorithms that are then applied globally. To my opinion, this situation has led to frequent overstatement about the performance of inversion algorithms applied to satellite ocean colour.*

**Response:** We thank the Reviewer for positive comments on our manuscript.

*The use of the Hydrolight RT code is a relevant choice, in particular because it allows Raman scattering and Chl fluorescence to be simulated. I would however recommend that the authors make clearer in the paper that modelling Chl fluorescence includes several assumptions so that any algorithm development using the part of the spectrum affected by Chl fluorescence will be largely depending on those assumptions. They allude to this on lines 577-578 (by mentioning that the quantum efficiency of Chl fluorescence was set to a default value) but a better description of this part and of the associated limitations should be included.*

**Response:** We agree and added two references showing that this parameter is variable (from about 0.01 to 0.05). The following sentence has been added in the revised manuscript (lines 568-570): "The quantum efficiency of chlorophyll-a fluorescence, which may exhibit significant variability (nearly 5-fold between about 0.01 and 0.05) in ocean waters (Maritorena et al., 2000; Morrison et al., 2003), was also set to its default value of 0.02 in the HydroLight code."

*The selection of possibly realistic combinations of absorption by phytoplankton, coloured dissolved organic matter (CDOM) and non-algal particles is important if users of the data set aim at retrieving these quantities. However, the authors know that RT in the ocean, for given boundary conditions, is entirely driven by only two inherent optical properties: the total absorption coefficient and the total volume scattering function (VSF). It does not matter for RT whether a given total absorption at a given wavelength is generated by various proportions of phytoplankton or water or anything else. The important parameters are therefore the ratio of particulate to molecular scattering (defining then the total VSF) and the ratio of total absorption to total scattering or total attenuation. Therefore, although I recognise that it is useful to see IOP distributions for components like phytoplankton and CDOM (Fig. 2), it would be equally useful to see the distributions of the parameters mentioned above. With what is presented, we cannot be sure that they are well covered.*

**Response:** We agree and following Reviewer's comment we have added in the revised manuscript the information about these ratios (lines 620-627), and a new figure (Figure 8 in the revised manuscript). This addition reads as follows "The radiative transfer is driven mainly by two ratios of IOPs which are the scattering to absorption ratio, $b(\lambda)/a(\lambda)$, which controls the number of scattering events (Morel and Gentili, 1991), and the molecular to total scattering ratio, $b_w(\lambda)/b(\lambda)$, which is the parameter controlling the weighted sum of the particle scattering and molecular scattering phase functions (Morel and Loisel, 1998; Loisel and Stramski, 2000). Figure 8 shows the distribution of these two ratios at 440 nm for the synthetic dataset. The $b_w(440)/b(440)$ and $b(440)/a(440)$ ratios range between about 0 and 0.2 and 0.5 and 10, respectively, which is consistent with previous models developed for Case-1 waters (Figs. 2 and 3 in Morel and Gentili, 1991; Fig. 2 in Morel and Loisel, 1998)."

*A few more detailed comments*
*Title: a data base of what? Maybe that could be said (Although the title is already quite long)*

**Response:** The title has been modified. We added "optical" in front of "database" which we think is adequate, especially that the database consists of numerous optical quantities including various IOPs, AOPs, and radiometric quantities, so it is impractical to attempt to create a title that would list all different categories of optical quantities included in the database.

*The entire section 2 is quite dense and not that easy to read. Some sub-headings might help better figure out the various steps followed in generating the IOP set. I guess many details could also be moved into a supplemental section, at least if the Journal allows it.*

**Response:** We agree and following Reviewer's comment have added four sub-headings (four subsections) in section 2 to facilitate reading. However, we believe it is important to have sufficiently detailed description of IOP dataset and how it was created because the database obtained through RT simulations depends critically on IOPs. Readers who may be less interested in the IOP-related details can skip this section with only minor implications to reading or understanding of RT-related section.

*Line 115: the use of Zhang et al to calculate the seawater scattering coefficient implies that a temperature and a salinity have been chosen. This should be indicated, and it should be made clear that this data set*

*cannot be used for freshwater, for instance. The manuscript title says "global ocean" but we know that not all users might be that careful.*

**Response:** We believe that this point is already clear in the manuscript but for further clarification we added the following sentence in section 5 (Summary): "Because of the use of absorption and scattering properties of pure seawater (assuming the salinity of 35‰) in the simulations, the present database cannot be used for applications to freshwater environments and also special caution should also be exercised for applications when water salinity is significantly less than 35‰ because of decrease in pure seawater scattering".

*Paragraph 199-214: the in-situ data bases that have been used should be listed and acknowledged. I know it can be a bit cumbersome because there are many, but a Table would make this efficiently. Citing papers that have previously used these data bases is not the correct way of doing it.*

**Response:** We agree that citing papers that have previously used these databases is not the best approach. We therefore removed such papers from the citation list but added several new citations to papers which describe the original collection of in situ data. Papers that describe the compilation of specific datasets and a list of general database sources used in our study are also provided. The revised description now reads (lines 214-220): "Many in situ data of IOP coefficients used in the present study were collected in previous studies (e.g., Reynolds et al., 2001; Babin et al., 2003; Loisel et al., 2007 ; Claustre et al., 2008; Huot et al., 2008; Stramski et al., 2008; Lubac et al., 2008 ; Loisel et al., 2009; Bricaud et al., 2010; Loisel et al., 2011; Antoine et al., 2011; Neukermans et al., 2012; Uitz et al., 2015; Neukermans et al., 2016; Reynolds et al., 2016; Aurin et al., 2018; Reynolds and Stramski, 2019; Stramski et al., 2019). Some data are described in publications devoted to compilation of various datasets (Valente et al., 2019; Casey et al., 2020) and are included in several databases (e.g., SeaBASS, CoastlOOC, BOUSSOLE, and GOCAD)."

As the Reviewer notes, because of the large number of individual datasets it would be quite cumbersome to provide a complete list, even in the form of a dedicated Table (several pages). As the majority of this assembled IOP dataset (e.g., coefficients for $a_g(\lambda)$, $a_{dg}(\lambda)$, $b_{bp}(\lambda)$) is not used in any quantitative way to develop the synthetic IOP dataset, but instead primarily serves as a means for comparison with the derived synthetic IOPs, we believe a detailed description of all data sources is unnecessary. With regards to measurements of $a_{ph}(\lambda)$, although these data are used to provide realistic spectral shapes of the phytoplankton absorption coefficient, as described in section 2.3 of the revised manuscript the vast majority of assembled data required additional modifications and refinements (e.g., spectral interpolation, extrapolation to the UV) to suit the purposes of our study. In addition, a large number of the initial assembled dataset of $a_{ph}(\lambda)$ (~50%) failed to meet certain criteria required for our use and were subsequently rejected and not included in the study. Finally, the distribution of spectral shapes obtained from these final accepted spectra was further modified to ensure a distribution representative of the global ocean. For these numerous reasons, we feel that a detailed description of all original sources of the in situ IOP dataset would, in addition to being impractical, have little to no value to the reader.

*The spectral values of the particulate backscattering coefficient resulting from summing up the phytoplankton and non-algal particle backscattering coefficients derived through Eqs. 8 and 11 should be displayed for some Chl values (or the spectral slope as a function of Chl could be displayed). That is an important parameter and it is unclear how it looks like.*

**Response:** Figure R2-1 below gives some insight into this question by showing $a_{ph}$ vs. the spectral slope of $b_{bp}$. We have not included this figure in the manuscript but added some relevant text (lines 489-492): "The spectral slope of $b_{bp}(\lambda)$, $\gamma$, where $b_{bp}(\lambda)$ is obtained as the sum of $b_{b\text{-}ph}(\lambda)$ and $b_{b\text{-}d}(\lambda)$, has a mean and

standard deviation of $1.10 \pm 0.34$, and exhibits a decreasing trend from oligotrophic (where $\gamma$ is around -2) to eutrophic waters (where the $b_{bp}(\lambda)$ spectrum is nearly flat). These results are in good agreement with previous studies (Morel and Maritorena, 2001; Loisel et al., 2006; Antoine et al., 2011)."

[Figure]

Fig. R2-1. Phytoplankton absorption coefficient, $a_{ph}(440)$, vs. the spectral slope of particulate backscattering coefficient for the new synthetic dataset.

*Pages 12-13: not sure I have got the rationale for the P1, 2, 3, 4 parameters and corresponding equations in Table 1. This should be better explained.*

**Response:** The parameters for $P_1$ and $P_2$ were chosen to match the relationships between $a_g(443)$ and $a_d(443)$, respectively, vs. $a_{ph}(443)$ that were observed with the in situ dataset used in this study. The $P_3$ and $P_4$ parameters were fixed based on the IOCCG (2006) formulations developed for both $b_{b\text{-}ph}$ and $b_{b\text{-}d}$, the two components of $b_{bp}$. The range of variability for the randomly-generated numbers involved in these $P$ formulations has been chosen to match both the in situ variability observed as well as the satellite distribution. Some explanatory text was added to the revised manuscript and a new column was added to Table 1 which provides references to the sources of these parameters.

*Pages 5-9: this is quite long for describing how absorption is modelled. And, there is conversely little about scattering (discussing the phase function only comes at page 14).*

**Response:** As absorption, specifically $a_{ph}(\lambda)$, is the main driver in determining the synthetic IOP dataset and subsequent RT calculations we believe that a detailed description is important for readers. Some additions in text and a new figure (Fig. 8) were added in the revised manuscript to enhance the description of modeled scattering properties in the IOP dataset, i.e., regarding (the spectral slope of $b_{bp}$ (lines 489-492), the $b/a$ and $b_w/b$ ratios (lines 620-627, new Fig. 8), and the particulate phase function (lines 473, 486-488).

*Fig. 9 would be advantageously completed by a similar plot of Rrs490/Rrs555 vs. Chl.*

**Response:** Following Reviewer's comment a new figure (Fig. 11) was added in the revised manuscript. This figure shows Chla as a function of the maximum band ratio used in SeaWiFS-specific OC5

algorithm (O'Reilly and Werdell, 2019). A short paragraph was added to describe this figure (lines 690-694).

*Another very important parameter to show would be the ratio of CDOM absorption to total non-water absorption vs. Chl. I guess what I am trying to say here is that a more comprehensive assessment of how this data set compares to previous ones and to established Case 1 water bio-optical models would be really useful.*

**Response:** We believe that our responses and several figures provided in response to Reviewer #1, especially Figs. R1-1, R1-4, and R1-5 with absorption data as well as an enhancement of Fig. 7 in the revised manuscript (the addition of panel c), address this comment by the Reviewer.

---

## Author Comment (AC3)

**Responses to Reviewer #3**

"A synthetic optical database generated by radiative transfer simulations in support of studies in ocean optics and optical remote sensing of the global ocean"

Hubert Loisel, Daniel Schaffer Ferreira Jorge, Rick A. Reynolds, and Dariusz Stramski

We appreciate the constructive comments by the Reviewer. Here we provide our detailed point-by-point responses and a description of any actions taken in regard to these comments. The Reviewer's comments are shown in italicized font; our responses follow each comment in normal font. Line numbers and figures indicated in our responses refer to the revised manuscript unless otherwise noted.

*The Authors present a new synthetic dataset for use in satellite ocean color algorithm development and refinement activities. In my opinion, this dataset improves upon previous versions (that also focused on the global ocean) in its attention to realistic distributions of bio-optical inputs, inclusion of various flavors of inelastic scattering, and the utility of including depth-resolved parameters as output. Overall, I found the manuscript to be mostly clear and otherwise very well written. I have no major concerns regarding its acceptance and publication. I have several minor editorial comments provided below that might be addressed.*

**Response:** We thank the Reviewer for positive comments on our manuscript.

*Line 49: Suggest providing units the first time a variable is introduced.*

**Response:** Done.

*Line 59: Suggest expanding the SeaBASS acronym and providing a URL or reference.*

**Response:** Done.

*Line 78: Suggest changing "the publicly available" to "a widely-used publicly available".*

**Response:** Done.

*Lines 83, 126, 131, 141, elsewhere (please check throughout the manuscript): which -> that*

**Response:** Checked and changes were made where appropriate.

*Line 93: By "driven" do you mean "described"?*

**Response:** Changed to "described by"

*Line 98: No "and" in the PACE acronym.*

**Response:** Corrected.

*Line 246: Suggest changing "a couple of" to "several".*

**Response:** Done.

*Lines 242-264: So, I think I understand what was done, but it took me several reads. The paragraph reads clumsily to me – wondering if there's a way to tighten it up (make it more concise and clear) and/or add a flow chart? Also, how many aph's were originally hyperspectral and how many fit into each of the two multispectral categories?*

**Response:** We agree that reading information related to creation of $a_{ph}(\lambda)$ dataset can be somewhat challenging because of various details involved in this process. However, we think we have done a reasonably good job to balance the general concept and various details underlying this task and we believe it is important to have sufficiently detailed description of IOP dataset and how it was created because the database obtained through RT simulations depends critically on IOPs. Readers who may be less interested in the IOP-related details can skip this section with only minor implications to reading or understanding of RT-related section.

For all final spectra that passed the described analysis and criteria (i.e., 2204 spectra), 593 measurements were originally hyperspectral, 65 spectra were created from spectra with 30 to 200 spectral bands, and the remaining 1546 spectra were created from measurements with a number of spectral bands lower than 30. Text relevant to this question was added to the revised manuscript (lines 272-277).

*Lines 275-277: I'm wondering if it's unequivocally ok to assume that just because two spectra match in the 400-750 nm region that they will also match in the 350-400 nm region. Could you comment on this?*

**Response:** While we recognize that this assumption is not necessarily satisfied (although it is possible that in many cases can be approximately satisfied), we consider it satisfactory for the purposes of extending some $a_{ph}$ spectra into the near-UV, especially that we do not expect significant or detrimental consequences of this assumption to the creation of synthetic database from RT simulations.

*Line 472: Suggest adding a reference for the 0.01 value.*

**Response:** The references of IOCCG (2006) and Loisel et al. (2007) have been added.

*Line 486: Suggest adding a reference for the 0.018 value.*

**Response:** The references of Petzold (1972) and Mobley (1994) have been added.

*Table 1: Please add references for the expressions or reiterate in the caption that these expressions are those used in previous studies (e.g., IOCCG 2006, Craig et al. 2020).*

**Response:** A new column was added to Table 1 in the revised manuscript that indicates references for each expression.

---

## Author Response (AR2)

***Q1:*** *In some places, chlorophyll is estimated from aph, but the 440 nm band is used, which is divided by 0.05582. This is gross and does not consider packaging effect. At a zero cost the power law formula from aph(660) could be used instead.*

**Response:** We agree that $a_{ph}$ in the red is a better proxy of Chla because this band is less affected by various accessory pigments and package effect than the blue absorption band. However, in these specific calculations this aspect is not critically important because the purpose is to generate a relatively large range of variability in one of the IOP coefficients, which is accomplished through the use of a random factor, and not to predict Chla.

***Q2:*** *The backscattering ratios of phytoplankton and non-algal particles have to be assumed. regarding the former, there is some data by Whitmire et al. (2010), so 0.01 has not to be assumed anymore. Also, that ratio can be related to other parameters such as the specific scattering or backscattering of phytoplankton.*

**Response:** The paragraph has been modified as follows: "where 0.01 is the value of backscattering ratio of phytoplankton, $\tilde{b}_{b-ph}$, assumed to be constant and independent of light wavelength in the present study (IOCCG, 2006; Loisel et al., 2007). Laboratory measurements performed on various phytoplankton cultures have shown, however, that, $\tilde{b}_{b-ph}$ can exhibit a slight spectral variation with the value at 442 nm ranging from 0.0035 to 0.029  (Whitmire et al., 2010)."

***Q3:*** *Fig. 12 is trivial and suits more a textbook than a paper with new findings. There are many ways in which the resulting AOPs can be presented. For instance, an ad-hoc classification in optical water types, and for each class the ternary plot of the absorption budget.*

**Response:** As mentioned in our previous response to Reviewer, the purpose of this figure is not to provide scientifically novel information but rather to illustrate the spectral and vertically-resolved (along the water column) optical information included in this new synthetic database. This can be useful to readers interested in this kind of optical data, especially that other commonly known synthetic optical databases (IOCCG, 2006; Craig et al., 2020) do not include data as a function of depth within the water column.